# MULTI-MODAL LATENT DIFFUSION

## ABSTRACT

Multi-modal data-sets are ubiquitous in modern applications, and multi-modal Variational Autoencoders are a popular family of models that aim to learn a joint representation of the different modalities. However, existing approaches suffer from a coherence–quality tradeoff, where models with good generation quality lack generative coherence across modalities, and vice versa. We discuss the limitations underlying the unsatisfactory performance of existing methods, to motivate the need for a different approach. We propose a novel method that uses a set of independently trained, uni-modal, deterministic autoencoders. Individual latent variables are concatenated into a common latent space, which is fed to a masked diffusion model to enable generative modeling. We also introduce a new multi-time training method to learn the conditional score network for multi-modal diffusion. Our methodology substantially outperforms competitors in both generation quality and coherence, as shown through an extensive experimental campaign.

## 1 INTRODUCTION

Multi-modal generative modelling is a crucial area of research in machine learning that aims to develop models capable of generating data according to multiple modalities, such as images, text, audio, and more. This is important because real-world observations are often captured in various forms, and combining multiple modalities describing the same information can be an invaluable asset. For instance, images and text can provide complementary information in describing an object, audio and video can capture different aspects of a scene. Multi-modal generative models can also help in tasks such as data augmentation (He et al., 2023; Azizi et al., 2023; Sariyildiz et al., 2023), missing modality imputation (Antelmi et al., 2019; Da Silva–Filarder et al., 2021; Zhang et al., 2023; Tran et al., 2017), and conditional generation (Huang et al., 2022; Lee et al., 2019b).

Multi-modal models have flourished over the past years and have seen a tremendous interest from academia and industry, especially in the content creation sector. Whereas most recent approaches focus on specialization, by considering text as primary input to be associated mainly to images (Rombach et al., 2022; Saharia et al., 2022; Ramesh et al., 2022; Tao et al., 2022; Wu et al., 2022; Nichol et al., 2022; Chang et al., 2023) and videos (Blattmann et al., 2023; Hong et al., 2023; Singer et al., 2022), in this work we target an established literature whose scope is more general, and in which all modalities are considered equally important. A large body of work rely on extensions of the Variational Autoencoder (VAE) (Kingma & Welling, 2014) to the multi-modal domain: initially interested in learning joint latent representation of multi-modal data, such works have mostly focused on generative modeling. Multi-modal generative models aim at *high-quality* data generation, as well as generative *coherence* across all modalities. These objectives apply to both joint generation of new data, and to conditional generation of missing modalities, given a disjoint set of available modalities.

In short, multi-modal VAEs rely on combinations of uni-modal VAEs, and the design space consists mainly in the way the uni-modal latent variables are combined, to construct the joint posterior distribution. Early work such as Wu & Goodman (2018) adopt a product of experts approach, whereas others Shi et al. (2019) consider a mixture of expert approach. Product-based models achieve high generative quality, but suffer in terms of both joint and conditional coherence. This was found to be due to experts mis-calibration issues (Shi et al., 2019; Sutter et al., 2021). On the other hand, mixture-based models produce coherent but qualitatively poor samples. A first attempt to address the so called **coherence–quality tradeoff** (Daunhawer et al., 2022) is represented by the mixture of product of experts approach (Sutter et al., 2021). However recent comparative studies (Daunhawer et al., 2022) show that none of the existing approaches fulfill both the generative quality

and coherence criteria. A variety of techniques aim at finding a better operating point, such as contrastive learning techniques (Shi et al., 2021), hierarchical schemes (Vasco et al., 2022), total correlation based calibration of single modality encoders (Hwang et al., 2021), or different training objectives Sutter et al. (2020). More recently, the work in (Palumbo et al., 2023) considers explicitly separated shared and private latent spaces to overcome the aforementioned limitations.

By expanding on results presented in (Daunhawer et al., 2022), in Section 2 we further investigate the tradeoff between generative coherence and quality, and argue that it is intrinsic to all variants of multi-modal VAEs. We indicate two root causes of such problem: latent variable collapse (Alemi et al., 2018; Dieng et al., 2019) and information loss due to mixture sub-sampling. To tackle these issues, in this work, we propose in Section 3 a new approach which uses a set of independent, uni-modal *deterministic* auto-encoders whose latent variables are simply concatenated in a joint latent variable. Joint and conditional generative capabilities are provided by an additional model that learns a probability density associated to the joint latent variable. We propose an extension of score-based diffusion models (Song et al., 2021b) to operate on the multi-modal latent space. We thus derive both forward and backward dynamics that are compatible with the multi-modal nature of the latent data. In section 4 we propose a novel method to train the multi-modal score network, such that it can both be used for joint and conditional generation. Our approach is based on a guidance mechanism, which we compare to alternatives. We label our approach Multi-modal Latent Diffusion (MLD).

Our experimental evaluation of MLD in Section 5 provides compelling evidence of the superiority of our approach for multi-modal generative modeling. We compare MLD to a large variety of VAE-based alternatives, on several real-life multi-modal data-sets, in terms of generative quality and both joint and conditional coherence. Our model outperforms alternatives in all possible scenarios, even those that are notoriously difficult because modalities might be only loosely correlated. Note that recent work also explore the joint generation of multiple modalities Ruan et al. (2023); Hu et al. (2023), but such approaches are application specific, e.g. text-to-image, and essentially only target two modalities. When relevant, we compare our method to additional recent alternatives to multi-modal diffusion (Bao et al., 2023; Wesego & Rooshenas, 2023), and show superior performance of MLD.

## 2 LIMITATIONS OF MULTI-MODAL VAES

In this work, we consider multi-modal VAEs (Wu & Goodman, 2018; Shi et al., 2019; Sutter et al., 2021; Palumbo et al., 2023) as the standard modeling approach to tackle both joint and conditional generation of multiple modalities. Our goal here is to motivate the need to go beyond such a standard approach, to overcome limitations that affect multi-modal VAEs, which result in a trade-off between generation quality and generative coherence (Daunhawer et al., 2022; Palumbo et al., 2023).

Consider the random variable $X = \{X^1, \ldots, X^M\} \sim p_D(x^1, \ldots, x^M)$, consisting in the set of $M$ of modalities sampled from the (unknown) multi-modal data distribution $p_D$. We indicate the marginal distribution of a single modality by $X^i \sim p_D^i(x^i)$ and the collection of a generic subset of modalities by $X^A \sim p_D^A(x^A)$, with $X^A \stackrel{\text{def}}{=} \{X^i\}_{i \in A}$, where $A \subset \{1, \ldots, M\}$ is a set of indexes. For example: given $A = \{1, 3, 5\}$, then $X^A = \{X^1, X^3, X^5\}$.

We begin by considering uni-modal VAEs as particular instances of the Markov chain $X \to Z \to \hat{X}$, where $Z$ is a latent variable and $\hat{X}$ is the generated variable. Models are specified by the two conditional distributions, called the encoder $Z \mid X=x \sim q_\psi(z \mid x)$, and the decoder $\hat{X} \mid Z=z \sim p_\theta(\hat{x} \mid z)$. Given a prior distribution $p_n(z)$, the objective is to define a generative model whose samples are distributed as closely as possible to the original data.

In the case of multi-modal VAEs, we consider the general family of Mixture of Product of Experts (MOPOE) (Sutter et al., 2021), which includes as particular cases many existing variants, such as Product of Experts (MVAE) (Wu & Goodman, 2018) and Mixture of Expert (MMVAE) (Shi et al., 2019). Formally, a collection of $K$ arbitrary subsets of modalities $S = \{A_1, \ldots A_K\}$, along with weighting coefficients $\omega_i \geq 0, \sum_{i=1}^{K} \omega_i = 1$, define the posterior $q_\psi(z \mid x) = \sum_i \omega_i q_{\psi^{A_i}}^i(z \mid x^{A_i})$, with $\psi = \{\psi^1, \ldots, \psi^K\}$. To lighten the notation, we use $q_{\psi^{A_i}}$ in place of $q_{\psi^{A_i}}^i$ noting that the various $q_{\psi^{A_i}}^i$ can have both different parameters $\psi^{A_i}$ and functional form. For example, in the MOPOE (Sutter et al., 2021) parametrization, we have: $q_{\psi^{A_i}}(z \mid x^{A_i}) = \prod_{j \in A_i} q_{\psi^j}(z \mid x^j)$. Our exposition is

more general and not limited to this assumption. The selection of the posterior can be understood as the result induced by the two step procedure where i) each subset of modalities $A_i$ is encoded into specific latent variables $Y_i \sim q_{\psi^{A_i}}(\cdot \,|\, x^{A_i})$ and ii) the latent variable $Z$ is obtained as $Z = Y_i$ with probability $\omega_i$. Optimization is performed w.r.t. the following evidence lower bound (ELBO) (Daunhawer et al., 2022; Sutter et al., 2021):

$$\mathcal{L} = \sum_i \omega_i \int p_D(x) q_{\psi^{A_i}}(z \,|\, x^{A_i}) \log p_\theta(x|z) - \log \frac{q_{\psi^{A_i}}(z \,|\, x^{A_i})}{p_n(z)} \mathrm{d}z\mathrm{d}x. \tag{1}$$

A well-known limitation called the latent collapse problem (Alemi et al., 2018; Dieng et al., 2019) affects the quality of latent variables $Z$. Consider the hypothetical case of arbitrary flexible encoders and decoders: then, posteriors with zero mutual information with respect to model inputs are valid maximizers of Equation (1). To prove this, it is sufficient to substitute the posteriors $q_{\psi^{A_i}}(z \,|\, x^{A_i}) = p_n(z)$ and $p_\theta(x|z) = p_D(x)$ into the Equation (1) to observe that the optimal value $\mathcal{L} = \int p_D(x) \log p_D(x) \mathrm{d}x$ is achieved (Alemi et al., 2018; Dieng et al., 2019). The problem of information loss is exacerbated in the case of multi-modal VAEs (Daunhawer et al., 2022). Intuitively, even if the encoders $q_{\psi^{A_i}}(z \,|\, x^{A_i})$ carry relevant information about their inputs $X^{A_i}$, step ii) of the multi-modal encoding procedure described above induces a further information bottleneck. A fraction $\omega_i$ of the time, the latent variable $Z$ will be a copy of $Y_i$, that only provides information about the subset $X^{A_i}$. No matter how good the encoding step is, the information about $X^{\{1,\ldots,M\}\backslash A}$ that is not contained in $X^{A_i}$ cannot be retrieved.

Furthermore, if the latent variable carries zero mutual information w.r.t. the multi-modal input, a coherent *conditional* generation of a set of modalities given others is impossible, since $\hat{X}^{A_1} \perp X^{A_2}$ for any generic sets $A_1, A_2$. While the factorization $p_\theta(x \,|\, z) = \prod_{i=1}^M p_{\theta^i}(x^i \,|\, z)$, $\theta = \{\theta_1, \ldots, \theta_M\}$ — where we use $p_{\theta^i}$ instead of $p_{\theta^i}^i$ to unclutter the notation — could enforce preservation of information and guarantee a better quality of the *jointly* generated data, in practice, the latent collapse phenomenon induces multi-modal VAEs to converge toward sub-optimal operating regime. When the posterior $q_\psi(z \,|\, x)$ collapses onto the uninformative prior $p_n(z)$, the ELBO in Equation (1) reduces to the sum of modality independent reconstruction terms $\sum_i \omega_i \sum_{j \in A_i} \int p_D^j(x^j) p_n(z) \left( \log p_{\theta^j}(x^j|z) \right) \mathrm{d}z\mathrm{d}x^j$.

In this case, flexible decoders can similarly ignore the latent variable and converge to the solution $p_{\theta^j}(x^j|z) = p_D^j(x^j)$ where, paradoxically, the quality of the approximation of the various marginal distributions is extremely high, while there is a complete lack of joint coherence.

General principles to avoid latent collapse consist in explicitly forcing the learning of informative encoders $q_\theta(z \,|\, x)$ via $\beta-$annealing of the Kullback-Leibler (KL) term in the ELBO and the reduction of the representational power of encoders and decoders. While $\beta-$annealing has been explored in the literature (Wu & Goodman, 2018) with limited improvements, reducing the flexibility of encoders/decoders clearly impacts the generation quality. Hence the presence of a trade-off: to improve coherence, the flexibility of encoders/decoders should be constrained, which in turns hurt generative quality. This trade-off has been recently addressed in the literature of multi-modal VAEs (Daunhawer et al., 2022; Palumbo et al., 2023), but our experimental results in Section 5 indicate that there is ample room for improvement, and that a new approach is truly needed.

## 3 OUR APPROACH: MULTI-MODAL LATENT DIFFUSION

We propose a new method for multi-modal generative modeling that, by design, does not suffer from the limitations discussed in Section 2. Our objective is to enable both high-quality and coherent joint/conditional data generation, using a simple design (see Appendix A for a schematic representation). As an overview, we use deterministic uni-modal autoencoders, whereby each modality $X^i$ is encoded through its encoder $e_{\psi^i}$, which is a short form for $e^i_{\psi^i}$, into the modality specific latent variable $Z^i$ and decoded into the corresponding $\hat{X}^i = d_{\theta^i}(Z^i)$. Our approach can be interpreted as a latent variable model where the different latent variables $Z^i$ are concatenated as $Z = [Z^1, \ldots, Z^M]$. This corresponds to the parametrization of the two conditional distributions as $q_\psi(z \,|\, x) = \prod_{i=1}^M \delta(z^i - e_{\psi^i}(x^i))$ and $p_\theta(\hat{x} \,|\, z) = \prod_{i=1}^M \delta(\hat{x}^i - d_{\theta^i}(z^i))$, respectively. Then, in place of an ELBO, we optimize the parameters of our autoencoders by minimizing the following sum of

modality specific losses:

$$\mathcal{L} = \sum_{i=1}^{M} \mathcal{L}_i, \quad \mathcal{L}_i = \int p_D^i(x^i) l^i(x^i - d_{\theta^i}(e_{\psi^i}(x^i))) \mathrm{d}x^i, \tag{2}$$

where $l^i$ can be any valid distance function, e.g, the square norm $\|\cdot\|^2$. Parameters $\psi^i, \theta^i$ are modality specific: then, minimization of Equation (2) corresponds to individual training of the different autoencoders. Since the mapping from input to latent is deterministic, there is no loss of information between $X$ and $Z$.[1] Moreover, this choice avoids any form of interference in the back-propagated gradients corresponding to the uni-modal reconstruction losses. Consequently gradient conflicts issues (Javaloy et al., 2022), where stronger modalities pollute weaker ones, are avoided.

To enable such a simple design to become a generative model, it is sufficient to generate samples from the induced latent distribution $Z \sim q_\psi(z) = \int p_D(x) q_\psi(z \,|\, x) \mathrm{d}x$ and decode them as $\hat{X} = d_\theta(Z) = [d_{\theta^1}(Z^1), \dots, d_{\theta^M}(Z^M)]$. To obtain such samples, we follow the two-stage procedure described in Loaiza-Ganem et al. (2022); Tran et al. (2021), where samples from the lower dimensional $q_\psi(z)$ are obtained through an appropriate generative model. We consider score-based diffusion models in latent space (Rombach et al., 2022; Vahdat et al., 2021) to solve this task, and call our approach Multi-modal Latent Diffusion (MLD). It may be helpful to clarify, at this point, that the two-stage training of MLD is carried out separately. Uni-modal deterministic autoencoders are pre-trained first, followed by the training of the score-based diffusion model, which is explained in more detail later.

To conclude the overview of our method, for joint data generation, one can sample from noise, perform backward diffusion, and then decode the generated multi-modal latent variable to obtain the corresponding data samples. For conditional data generation, given one modality, the reverse diffusion is guided by this modality, while the other modalities are generated by sampling from noise. The generated latent variable is then decoded to obtain data samples of the missing modality.

### 3.1 Joint and Conditional Multi-modal Latent Diffusion Processes

In the first stage of our method, the deterministic encoders project the input modalities $X^i$ into the corresponding latent spaces $Z^i$. This transformation induces a distribution $q_\psi(z)$ for the latent variable $Z = [Z^1, \dots, Z^M]$, resulting from the concatenation of uni-modal latent variables.

**Joint generation.** To generate a new sample for all modalities we use a simple score-based diffusion model in latent space (Sohl-Dickstein et al., 2015; Song et al., 2021b; Vahdat et al., 2021; Loaiza-Ganem et al., 2022; Tran et al., 2021). This requires reversing a stochastic noising process, starting from a simple, Gaussian distribution. Formally, the noising process is defined by a Stochastic Differential Equation (SDE) of the form:

$$\mathrm{d}R_t = \alpha(t) R_t \mathrm{d}t + g(t) \mathrm{d}W_t, \ \ R_0 \sim q(r, 0), \tag{3}$$

where $\alpha(t) R_t$ and $g(t)$ are the drift and diffusion terms, respectively, and $W_t$ is a Wiener process. The time-varying probability density $q(r, t)$ of the stochastic process at time $t \in [0, T]$, where $T$ is finite, satisfies the Fokker-Planck equation (Oksendal, 2013), with initial conditions $q(r, 0)$. We assume uniqueness and existence of a stationary distribution $\rho(r)$ for the process Equation (3).[2] The forward diffusion dynamics depend on the initial conditions $R_0 \sim q(r, 0)$. We consider $R_0 = Z$ to be the initial condition for the diffusion process, which is equivalent to $q(r, 0) = q_\psi(r)$. Under loose conditions (Anderson, 1982), a time-reversed stochastic process exists, with a new SDE of the form:

$$\mathrm{d}R_t = \left(-\alpha(T-t) R_t + g^2(T-t) \nabla \log(q(R_t, T-t))\right) \mathrm{d}t + g(T-t) \mathrm{d}W_t, \ \ R_0 \sim q(r, T), \tag{4}$$

indicating that, in principle, simulation of Equation (4) allows to generate samples from the desired distribution $q(r, 0)$. In practice, we use a **parametric score network** $s_\chi(r, t)$ to approximate the true score function, and we approximate $q(r, T)$ with the stationary distribution $\rho(r)$. Indeed, the generated data distribution $q(r, 0)$ is close (in KL sense) to the true density as described by Song et al. (2021a); Franzese et al. (2023):

$$\mathrm{KL}[q_\psi(r) \,|\, q(r, 0)] \leq \frac{1}{2} \int_0^T g^2(t) \mathbb{E}[\|s_\chi(R_t, t) - \nabla \log q(R_t, t)\|^2] \mathrm{d}t + KL[q(r, T) \| \rho(r)], \tag{5}$$

---

[1]Since the measures are not absolutely continuous w.r.t the Lebesgue measure, mutual information is $+\infty$.
[2]This is not necessary for the validity of the method Song et al. (2021a)

where the first term on the r.h.s is referred to as score-matching objective, and is the loss over which the score network is optimized, and the second is a vanishing term for $T \to \infty$.

To conclude, joint generation of all modalities is achieved through the simulation of the reverse-time SDE in Equation (4), followed by a simple decoding procedure. Indeed, optimally trained decoders (achieving zero in Equation (2)) can be used to transform $Z \sim q_\psi(z)$ into samples from $\int p_\theta(x \mid z) q_\psi(z) \mathrm{d}z = p_D(x)$.

**Conditional generation.** Given a generic partition of all modalities into non overlapping sets $A_1 \cup A_2$, where $A_2 = (\{1, \ldots, M\} \setminus A_1)$, conditional generation requires samples from the conditional distribution $q_\psi(z^{A_1} \mid z^{A_2})$, which are based on *masked* forward and backward diffusion processes.

Given conditioning latent modalities $z^{A_2}$, we consider a modified forward diffusion process with initial conditions $R_0 = \mathcal{C}(R_0^{A_1}, R_0^{A_2})$, with $R_0^{A_1} \sim q_\psi(r^{A_1} \mid z^{A_2}), R_0^{A_2} = z^{A_2}$. The composition operation $\mathcal{C}(\cdot)$ concatenates generated ($R^{A_1}$) and conditioning latents ($z^{A_2}$). As an illustration, consider $A_1 = \{1, 3, 5\}$, such that $X^{A_1} = \{X^1, X^3, X^5\}$, and $A_2 = \{2, 4, 6\}$ such that $X^{A_2} = \{X^2, X^4, X^6\}$. Then, $R_0 = \mathcal{C}(R_0^{A_1}, R_0^{A_2}) = \mathcal{C}(R_0^{A_1}, z^{A_2}) = [R_0^1, z^2, R_0^3, z^4, R_0^5, z^6]$.

More formally, we define the masked forward diffusion SDE:

$$\mathrm{d}R_t = m(A_1) \odot [\alpha(t)R_t \mathrm{d}t + g(t)\mathrm{d}W_t], \quad q(r, 0) = q_\psi(r^{A_1} \mid z^{A_2})\delta(r^{A_2} - z^{A_2}). \tag{6}$$

The mask $m(A_1)$ contains $M$ vectors $u^i$, one per modality, and with the corresponding cardinality. If modality $j \in A_1$, then $u^j = \mathbf{1}$, otherwise $u^j = \mathbf{0}$. Then, the effect of masking is to "freeze" throughout the diffusion process the part of the random variable $R_t$ corresponding to the conditioning latent modalities $z^{A_2}$. We naturally associate to this modified forward process the conditional time varying density $q(r, t \mid z^{A_2}) = q(r^{A_1}, t \mid z^{A_2})\delta(r^{A_2} - z^{A_2})$.

To sample from $q_\psi(z^{A_1} \mid z^{A_2})$, we derive the reverse-time dynamics of Equation (6) as follows:

$$\mathrm{d}R_t = m(A_1) \odot \left[\left(-\alpha(T-t)R_t + g^2(T-t)\nabla \log\big(q(R_t, T-t \mid z^{A_2})\big)\right)\mathrm{d}t + g(T-t)\mathrm{d}W_t\right], \tag{7}$$

with initial conditions $R_0 = \mathcal{C}(R_0^{A_1}, z^{A_2})$ and $R_0^{A_1} \sim q(r^{A_1}, T \mid z^{A_2})$. Then, we approximate $q(r^{A_1}, T \mid z^{A_2})$ by its corresponding steady state distribution $\rho(r^{A_1})$, and the true (conditional) score function $\nabla \log\big(q(r, t \mid z^{A_2})\big)$ by a conditional score network $s_\chi(r^{A_1}, t \mid z^{A_2})$.

## 4 GUIDANCE MECHANISMS TO LEARN THE CONDITIONAL SCORE NETWORK

A correctly optimized score network $s_\chi(r, t)$ allows, through simulation of Equation (4), to obtain samples from the joint distribution $q_\psi(z)$. Similarly, a *conditional* score network $s_\chi(r^{A_1}, t \mid z^{A_2})$ allows, through the simulation of Equation (7), to sample from $q_\psi(z^{A_1} \mid z^{A_2})$. In Section 4.1 we extend guidance mechanisms used in classical diffusion models to allow multi-modal conditional generation. A naïve alternative is to rely on the unconditional score network $s_\chi(r, t)$ for the conditional generation task, by casting it as an *in-painting* objective. Intuitively, any missing modality could be recovered in the same way as a uni-modal diffusion model can recover masked information. In Section 4.2 we discuss the implicit assumptions underlying in-painting from an information theoretic perspective, and argue that, in the context of multi-modal data, such assumptions are difficult to satisfy. Our intuition is corroborated by ample empirical evidence, where our method consistently outperform alternatives.

### 4.1 MULTI-TIME DIFFUSION

We propose a modification to the classifier-free guidance technique (Ho & Salimans, 2022) to learn a score network that can generate conditional and unconditional samples from any subset of modalities. Instead of training a separate score network for each possible combination of conditional modalities, which is computationally infeasible, we use a single architecture that accepts all modalities as inputs and a *multi-time vector* $\tau = [t_1, \ldots, t_M]$. The multi-time vector serves two purposes: it is both a conditioning signal and the time at which we observe the diffusion process.

**Training:** learning the conditional score network relies on randomization. As discussed in Section 3.1, we consider an arbitrary partitioning of all modalities in two disjoint sets, $A_1$ and $A_2$. The set $A_2$

contains randomly selected conditioning modalities, while the remaining modalities belong to set $A_1$. Then, during training, the parametric score network estimates $\nabla \log\big(q(r, t \,|\, z^{A_2})\big)$, whereby the set $A_2$ is randomly chosen at every step. This is achieved by the *masked diffusion process* from Equation (6), which only diffuses modalities in $A_1$. More formally, the score network input is $R_t = \mathcal{C}(R_t^{A_1}, Z^{A_2})$, along with a multi-time vector $\tau(A_1, t) = t\,[\mathbb{1}(1 \in A_1), \dots, \mathbb{1}(M \in A_1)]$. As a follow-up of the example in Section 3.1, given $A_1 = \{1, 3, 5\}$, such that $X^{A_1} = \{X^1, X^3, X^5\}$, and $A_2 = \{2, 4, 6\}$ such that $X^{A_2} = \{X^2, X^4, X^6\}$, then, $\tau(A_1, t) = [t, 0, t, 0, t, 0]$.

More precisely, the algorithm for the multi-time diffusion training (see A for the pseudo-code) proceeds as follows. At each step, a set of conditioning modalities $A_2$ is sampled from a predefined distribution $\nu$, where $\nu(\emptyset) \stackrel{\text{def}}{=} \Pr(A_2 = \emptyset) = d$, and $\nu(U) \stackrel{\text{def}}{=} \Pr(A_2 = U) = {(1-d)}/{(2^M - 1)}$ with $U \in \mathcal{P}(\{1, \dots, M\}) \setminus \emptyset$, where $\mathcal{P}(\{1, \dots, M\})$ is the powerset of all modalities. The corresponding set $A_1$ and mask $m(A_1)$ are constructed, and a sample $X$ is drawn from the training data-set. The corresponding latent variables $Z^{A_1} = \{e_\psi^i(X^i)\}_{i \in A_1}$ and $Z^{A_2} = \{e_\psi^i(X^i)\}_{i \in A_2}$ are computed using the pre-trained encoders, and a diffusion process starting from $R_0 = \mathcal{C}(Z^{A_1}, Z^{A_2})$ is simulated for a randomly chosen diffusion time $t$, using the conditional forward SDE with the mask $m(A_1)$. The score network is then fed the current state $R_t$ and multi-time vector $\tau(A_1, t)$, and the difference between the score network's prediction and the true score is computed, applying the mask $m(A_1)$. The score network parameters are updated using stochastic gradient descent, and this process is repeated for a total of $L$ training steps. Clearly, when $A_2 = \emptyset$, training proceeds as for an un-masked diffusion process, since the mask $m(A_1)$ allows all latent variables to be diffused.

**Conditional generation:** any valid numerical integration scheme for Equation (7) can be used for conditional sampling (see A for an implementation using the Euler-Maruyama integrator). First, conditioning modalities in the set $A_2$ are encoded into the corresponding latent variables $z^{A_2} = \{e^j(x^j)\}_{j \in A_2}$. Then, numerical integration is performed with step-size $\Delta t = T/N$, starting from the initial conditions $R_0 = \mathcal{C}(R_0^{A_1}, z^{A_2})$, with $R_0^{A_1} \sim \rho(r^{A_1})$. At each integration step, the score network $s_\chi$ is fed the current state of the process and the multi-time vector $\tau(A_1, \cdot)$. Before updating the state, the masking is applied. Finally, the generated modalities are obtained thanks to the decoders as $\hat{X}^{A_1} = \{d_\theta^j(R_T^j)\}_{j \in A_1}$. Inference time conditional generation is not randomized: conditioning modalities are the ones that are available, whereas the remaining are the ones we wish to generate.

Any-to-any multi-modality has been recently studied through the composition of modality-specific diffusion models (Tang et al., 2023), by designing cross-attention and training procedures that allow arbitrary conditional generation. The work by Tang et al. (2023) relies on latent interpolation of input modalities, which is akin to mixture models, and uses it as conditioning signal for individual diffusion models. This is substantially different from the joint nature of the multi-modal latent diffusion we present in our work: instead of forcing entanglement through cross-attention between score networks, our model relies on joint diffusion process, whereby modalities naturally co-evolve according to the diffusion process. Another recent work (Wu et al., 2023) targets multi-modal conversational agents, whereby the strong, underlying assumption is to consider one modality, i.e., text, as a guide for the alignment and generation of other modalities. Even if conversational objectives are orthogonal to our work, techniques akin to instruction following for cross-generation, are an interesting illustration of the powerful capabilities of in-context learning of LLMs (Xie et al., 2022; Min et al., 2022).

### 4.2 IN-PAINTING AND ITS IMPLICIT ASSUMPTIONS

Under certain assumptions, given an unconditional score network $s_\chi(r, t)$ that approximates the true score $\nabla \log q(r, t)$, it is possible to obtain a conditional score network $s_\chi(r^{A_1}, t \,|\, z^{A_2})$, to approximate $\nabla \log q(r^{A_1}, t \,|\, z^{A_2})$. We start by observing the equality:

$$q(r^{A_1}, t \,|\, z^{A_2}) = \int q(\mathcal{C}(r^{A_1}, r^{A_2}), t \,|\, z^{A_2})\, \mathrm{d}r^{A_2} = \int \frac{q(z^{A_2} \,|\, \mathcal{C}(r^{A_1}, r^{A_2}), t)}{q_\psi(z^{A_2})} q(\mathcal{C}(r^{A_1}, r^{A_2}), t)\, \mathrm{d}r^{A_2},$$

(8)

where, with a slight abuse of notation, we indicate with $q(z^{A_2} \,|\, \mathcal{C}(r^{A_1}, r^{A_2}), t)$ the density associated to the event: the portion corresponding to $A_2$ of the latent variable $Z$ is equal to $z^{A_2}$ given that the whole diffused latent $R_t$ at time $t$, is equal to $\mathcal{C}(r^{A_1}, r^{A_2})$. In the literature, the quantity $q(z^{A_2} \,|\, \mathcal{C}(r^{A_1}, r^{A_2}), t)$ is typically approximated by dropping its dependency on $r^{A_1}$. This approxima-

tion can be used to manipulate Equation (8) as $q(r^{A_1}, t \mid z^{A_2}) \simeq \int q(r^{A_2}, t \mid z^{A_2}) q(r^{A_1}, t \mid r^{A_2}, t) \, \mathrm{d}r$. Further Monte-Carlo approximations (Song et al., 2021b; Lugmayr et al., 2022) of the integral allow implementation of a practical scheme, where an approximate conditional score network is used to generate conditional samples. This approach, known in the literature as *in-painting*, provides high quality results in several *uni-modal* application domains (Song et al., 2021b; Lugmayr et al., 2022).

The KL divergence between $q(z^{A_2} \mid \mathcal{C}(r^{A_1}, r^{A_2}), t)$ and $q(z^{A_2} \mid r^{A_2}, t)$ quantifies, fixing $r^{A_1}, r^{A_2}$, the discrepancy between the true and approximated conditional probabilities. Similarly, the expected KL divergence $\Delta = \int q(r, t) \mathrm{KL}[q(z^{A_2} \mid \mathcal{C}(r^{A_1}, r^{A_2}), t) \mid \mid q(z^{A_2} \mid r^{A_2}, t)] \mathrm{d}r$, provides information about the average discrepancy. Simple manipulations allow to recast this as a discrepancy in terms of mutual information $\Delta = I(Z^{A_2}; R_t^{A_1}, R_t^{A_2}) - I(Z^{A_2}; R_t^{A_2})$. Information about $Z^{A_2}$ is contained in $R_t^{A_2}$, as the latter is the result of a diffusion with the former as initial conditions, corresponding to the Markov chain $R_t^{A_2} \to Z^{A_2}$, and in $R_t^{A_1}$ through the Markov chain $Z^{A_2} \to Z^{A_1} \to R_t^{A_1}$. The positive quantity $\Delta$ is close to zero whenever the rate of loss of information w.r.t initial conditions is similar for the two subsets $A_1, A_2$. In other terms, $\Delta \simeq 0$ whenever out of the whole $R_t$, the portion $R_t^{A_2}$ is a sufficient statistic for $Z^{A_2}$.

The assumptions underlying the approximation are in general not valid in the case of multi-modal learning, where the robustness to stochastic perturbations of latent variables corresponding to the various modalities can vary greatly. Our claim are supported empirically by an ample analysis on real data in B, where we show that multi-time diffusion approach consistently outperforms in-painting.

## 5 EXPERIMENTS

We compare our method MLD to MVAE Wu & Goodman (2018), MMVAE Shi et al. (2019), MOPOE Sutter et al. (2021), Hierarchical Genertive Model (NEXUS) Vasco et al. (2022) and Multi-view Total Correlation Autoencoder (MVTCAE) Hwang et al. (2021), MMVAE+ Palumbo et al. (2023) re-implementing competitors in the same code base as our method, and selecting their best hyper-parameters (as indicated by the authors). For fair comparison, we use the same encoder/decoder architecture for all the models. For MLD, the score network is implemented using a simple stacked multilayer perceptron (MLP) with skip connections (see A for more details).

**Evaluation metrics.** *Coherence* is measured as in Shi et al. (2019); Sutter et al. (2021); Palumbo et al. (2023), using pre-trained classifiers on the generated data and checking the consistency of their outputs. *Generative quality* is computed using Fréchet Inception Distance (FID) Heusel et al. (2017) and Fréchet Audio Distance (FAD) Kilgour et al. (2019) scores for images and audio respectively. Full details on the metrics are included in C. All results are averaged over 5 seeds (We report standard deviation in E).

**Results.** Overall, MLD largely outperforms alternatives from the literature, **both** in terms of coherence and generative quality. VAE-based models suffer from a coherence–quality trad-off and modality collapse for highly heterogeneous data-sets. We proceed to show this on several standard benchmarks from the multi-modal VAE-based literature (see C for details on the data-sets).

The first data-set we consider is **MNIST-SVHN** ((Shi et al., 2019)), where the two modalities differ in complexity. High variability, noise and ambiguity makes attaining good coherence for the SVHN modality a challenging task. Overall, MLD outperforms all VAE-based alternatives in terms of coherency, especially in terms of joint generation and conditional generation of MNIST given SVHN, see Table 1. Mixture models (MMVAE, MOPOE) suffer from modality collapse (poor SVHN generation), whereas product of experts (MVAE, MVTCAE) generate better quality samples at the expense of SVHN to MNIST conditional coherence. Joint generation is poor for all VAE models. Interestingly, these models also fail at SVHN self-reconstruction which we discuss in E. MLD achieves the best performance also in terms of generation quality, as confirmed also by qualitative results (Figure 1) showing for example how MLD conditionally generates multiple SVHN digits within one sample, given the input MNIST image, whereas other methods fail to do so.

The Multi-modal Handwritten Digits data-set (**MHD**) (Vasco et al., 2022) contains gray-scale digit images, motion trajectory of the hand writing and sounds of the spoken digits. In our experiments, we do not use the label as a forth modality. While digit image and trajectory share a good amount of information, the sound modality contains a lot more of modality specific variation. Consequently,

Table 1: Generation coherence and quality for **MNIST-SVHN** ( M :MNIST, S: SVHN). The generation quality is measured in terms of Fréchet Modality Distance (FMD) for MNIST and FID for SVHN.

| Models | Coherence (%↑) | | | Quality (↓) | | | |
|---|---|---|---|---|---|---|---|
| | Joint | M → S | S → M | Joint(M) | Joint(S) | M → S | S → M |
| MVAE | 38.19 | 48.21 | 28.57 | 13.34 | 68.9 | 68.0 | 13.66 |
| MMVAE | 37.82 | 11.72 | 67.55 | 25.89 | 146.82 | 393.33 | 53.37 |
| MOPOE | 39.93 | 12.27 | 68.82 | 20.11 | 129.2 | 373.73 | 43.34 |
| NEXUS | 40.0 | 16.68 | 70.67 | 13.84 | 98.13 | 281.28 | 53.41 |
| MVTCAE | 48.78 | **81.97** | 49.78 | 12.98 | **52.92** | 69.48 | 13.55 |
| MMVAE+ | 17.64 | 13.23 | 29.69 | 26.60 | 121.77 | 240.90 | 35.11 |
| MMVAE+(K=10) | 41.59 | 55.3 | 56.41 | 19.05 | 67.13 | 75.9 | 18.16 |
| **MLD (ours)** | **85.22** | 83.79 | **79.13** | **3.93** | 56.36 | **57.2** | **3.67** |

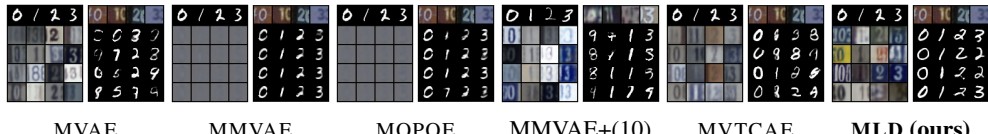

| MVAE | MMVAE | MOPOE | MMVAE+(10) | MVTCAE | **MLD (ours)** |

Figure 1: Qualitative results for **MNIST-SVHN**. For each model we report: MNIST to SVHN conditional generation in the left, SVHN to MNIST conditional generation in the right.

conditional generation involving the sound modality, along with joint generation, are challenging tasks. Coherency-wise (Table 2) MLD outperforms all the competitors where the biggest difference is seen in joint and sound to other modalities generation (in the latter task MVTCAE performs better than other competitors but is still worse than MLD). MLD dominates alternatives also in terms of generation quality (Table 3). This is true both for image, sound modalities, for which some VAE-based models suffer in producing high quality results, demonstrating the limitation of these methods in handling highly heterogeneous modalities. MLD, in the other hand, achieves high generation quality for all modalities, possibly due to the independent training of the autoencoders avoiding interference.

Table 2: Generation Coherence (%) for **MHD** (Higher is better). Line above refer to the generated modality while the observed modalities subset are presented below.

| Models | Joint | I (Image) | | | T (Trajectory) | | | S (Sound) | | |
|---|---|---|---|---|---|---|---|---|---|---|
| | | T | S | T,S | I | S | I,S | I | T | I,T |
| MVAE | 37.77 | 11.68 | 26.46 | 28.4 | 95.55 | 26.66 | 96.58 | 58.87 | 10.76 | 58.16 |
| MMVAE | 34.78 | **99.7** | 69.69 | 84.74 | 99.3 | 85.46 | 92.39 | 49.95 | 50.14 | 50.17 |
| MOPOE | 48.84 | 99.64 | 68.67 | 99.69 | 99.28 | 87.42 | 99.35 | 50.73 | 51.5 | 56.97 |
| NEXUS | 26.56 | 94.58 | 83.1 | 95.27 | 88.51 | 76.82 | 93.27 | 70.06 | 75.84 | 89.48 |
| MVTCAE | 42.28 | 99.54 | 72.05 | 99.63 | 99.22 | 72.03 | 99.39 | 92.58 | 93.07 | 94.78 |
| MMVAE+ | 41.67 | 98.05 | 84.16 | 91.88₊ | 97.47 | 81.16 | 89.31 | 64.34 | 65.42 | 64.88 |
| MMVAE+(k=10) | 42.60 | 99.44 | **89.75** | 94.7 | 99.44 | **89.58** | 95.01 | 87.15 | 87.99 | 87.57 |
| **MLD (ours)** | **98.34** | 99.45 | 88.91 | **99.88** | **99.58** | 88.92 | **99.91** | **97.63** | **97.7** | **98.01** |

The **POLYMNIST** data-set (Sutter et al., 2021) consists of 5 modalities synthetically generated by using MNIST digits and varying the background images. The homogeneous nature of the modalities is expected to mitigate gradient conflict issues in VAE-based models, and consequently reduce modality collapse. However, MLD still outperforms all alternatives, as shown Figure 2. Concerning generation coherence, MLD achieves the best performance in all cases with the single exception of a single observed modality. On the qualitative performance side, not only MLD is superior to alternatives, but its results are stable when more modalities are considered, a capability that not all competitors share.

Finally, we explore the Caltech Birds **CUB** (Shi et al., 2019) data-set, following the same experimentation protocol in Daunhawer et al. (2022) by using real bird images (instead of ResNet-features as in Shi et al. (2019)). Figure 3 presents qualitative results for caption to image conditional generation. MLD is the only model capable of generating bird images with convincing coherence. Clearly, none of the VAE-based methods is able to achieve sufficient caption to image conditional generation quality using the same simple autoencoder architecture. Note that an image autoencoder with larger capacity improves considerably MLD generative performance, suggesting that careful engineering applied to modality specific autoencoders is a promising avenue for future work. We report quantitative

Table 3: Generation quality for **MHD** in terms of FMD for image and trajectory modalities and FAD for the sound modality (Lower is better).

| Models | I (Image) | | | | T (Trajectory) | | | | S (Sound) | | | |
|---|---|---|---|---|---|---|---|---|---|---|---|---|
| | Joint | T | S | T,S | Joint | I | S | I,S | Joint | I | T | I,T |
| MVAE | 94.9 | 93.73 | 92.55 | 91.08 | 39.51 | 20.42 | 38.77 | 19.25 | 14.14 | 14.13 | 14.08 | 14.17 |
| MMVAE | 224.01 | 22.6 | 789.12 | 170.41 | 16.52 | **0.5** | 30.39 | 6.07 | 22.8 | 22.61 | 23.72 | 23.01 |
| MOPOE | 147.81 | 16.29 | 838.38 | 15.89 | 13.92 | 0.52 | 33.38 | **0.53** | 18.53 | 24.11 | 24.1 | 23.93 |
| NEXUS | 281.76 | 116.65 | 282.34 | 117.24 | 18.59 | 6.67 | 33.01 | 7.54 | 13.99 | 19.52 | 18.71 | 16.3 |
| MVTCAE | 121.85 | 5.34 | 54.57 | 3.16 | 19.49 | 0.62 | 13.65 | 0.75 | 15.88 | 14.22 | 14.02 | 13.96 |
| MMVAE+ | 97.19 | 2.80 | 128.56 | 114.3 | 22.37 | 1.21 | 21.74 | 15.2 | 16.12 | 17.31 | 17.92 | 17.56 |
| MMVAE+(K=10) | 85.98 | 1.83 | 70.72 | 62.43 | 21.10 | 1.38 | 8.52 | 7.22 | 14.58 | 14.33 | 14.34 | 14.32 |
| MLD | **7.98** | **1.7** | **4.54** | **1.84** | **3.18** | 0.83 | **2.07** | 0.6 | **2.39** | **2.31** | **2.33** | **2.29** |

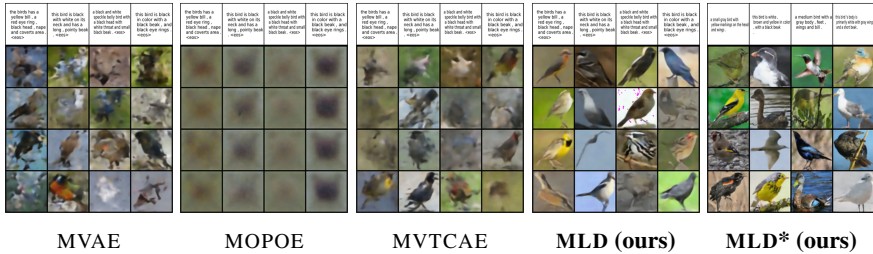

Figure 2: Results for **POLYMNIST** data-set. *Left*: a comparison of the generative coherence (% ↑) and quality in terms of FID (↓) as a function of the number of inputs. We report the average performance following the leave-one-out strategy (see C). *Right*: are qualitative results for the joint generation of the 5 modalities.

results in E, where we show generation quality FID metric. Due to the unavailability of the labels in this data-set, coherence evaluation as with the previous data-sets is not possible. We then resort to CLIP-Score (CLIP-S) Hessel et al. (2021) an image-captioning metric, that, despite its limitations for the considered data-set Kim et al. (2022), shows that MLD outperforms competitors.

# 6 CONCLUSION AND LIMITATIONS

We have presented a new multi-modal generative model, Multimodal Latent Diffusion (MLD), to address the well-known coherence–quality tradeoff that is inherent in existing multi-modal VAE-based models. MLD uses a set of independently trained, uni-modal, deterministic autoencoders. Generative properties of our model stem from a masked diffusion process that operates on latent variables. We also developed a new multi-time training method to learn the conditional score network for multi-modal diffusion. An extensive experimental campaign on various real-life data-sets, provided compelling evidence on the effectiveness of MLD for multi-modal generative modeling. In all scenarios, including cases with loosely correlated modalities and high-resolution datasets, MLD consistently outperformed the alternatives from the state-of-the-art.

Figure 3: Qualitative results on **CUB** data-set. Caption used as condition to generate the bird images. **MLD**\* denotes the version of our method using a powerful image autoencoder.

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

# A APPENDIX

## MULTI-MODAL LATENT DIFFUSION — SUPPLEMENTARY MATERIAL

## A DIFFUSION IN THE MULTIMODAL LATENT SPACE

In this section, we provide additional technical details of MLD. We first discuss a naive approach based on *In-painting* which uses only unconditional score network for both joint and conditional generation. We also discuss alternative training scheme based on a work from the caption-text translation literature Bao et al. (2023). Finally, we provide extra technical details for the score network architecture and sampling technique.

### A.1 MODALITIES AUTO-ENCODERS

Each deterministic autoencoders used in the first stage of MLD uses a vector latent space with no size constraints. Instead, VAE-based models, generally require the latent space of each individual VAE to be exactly of the same size, to allow the definition of a joint latent space.

In our approach, before concatenation, the modality-specific latent spaces are *normalized* by element-wise mean and standard deviation. In practice, we use the statistics retrieved from the first training batch, which we found sufficient to gain sufficient statistical confidence. This operation allows the harmonization of different modality-specific latent spaces and, therefore, facilitate the learning of a joint score network.

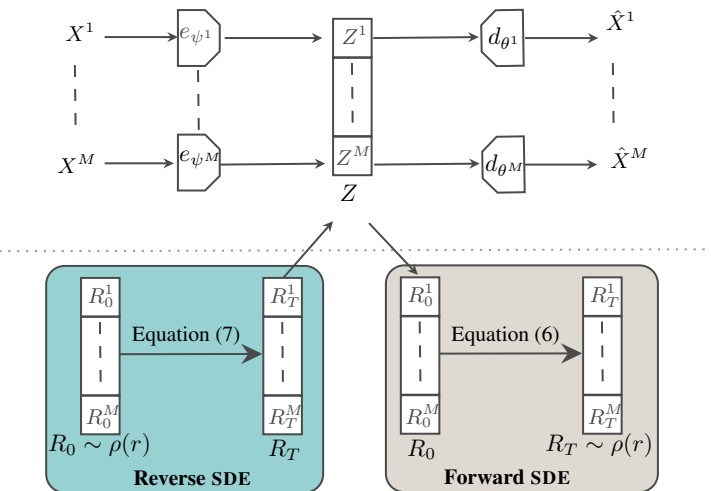

Figure 4: Multi-modal Latent Diffusion. Two-stage model involving: **Top:** deterministic, modality-specific encoder/decoders, **Bottom:** score-based diffusion model on the concatenated latent spaces.

### A.2 MULTI-MODAL DIFFUSION SDE

In Section 3, we presented our multi-modal latent diffusion process allowing multi-modal joint and conditional generation. The role of the SDE is to gradually add noise to the data, perturbing its structure until attaining a noise distribution. In this work, we consider Variance preserving SDE (VPSDE) Song et al. (2021b). In this framework we have : $\rho(r) \sim \mathcal{N}(0; I)$, $\alpha(t) = -\frac{1}{2}\beta(t)$ and $g(t) = \sqrt{\beta(t)}$, where $\beta(t) = \beta_{min} + t(\beta_{max} - \beta_{min})$. Following (Ho et al., 2020; Song et al., 2021b), we set $\beta_{min} = 0.1$ and $\beta_{max} = 20$. With this configuration and by substitution of Equation (3), we obtain the following forward SDE:

$$dR_t = -\frac{1}{2}\beta(t)R_t dt + \sqrt{\beta(t)}dW_t, \qquad t \in [0, T]. \tag{9}$$

The corresponding perturbation kernel is given by :

$$q(r|z,t) = \mathcal{N}(r; e^{-\frac{1}{4}t^2(\beta_{max}-\beta_{min})-\frac{1}{2}t\beta_{min}}z, (1 - e^{-\frac{1}{2}t^2(\beta_{max}-\beta_{min})-t\beta_{min}})\mathbf{I}). \tag{10}$$

The marginal score $\nabla \log q(R_t, t)$ is approximated by a score network $s_\chi(R_t, t)$ whose parameters $\chi$ can be optimized by minimizing the ELBO in Equation (5), where we found that using the same re-scaling as in Song et al. (2021b) is more stable.

The reverse process is described by a different SDE (Equation (4)). When using a variance-preserving SDE, Equation (4) specializes in:

$$dR_t = \left[\frac{1}{2}\beta(T-t)R_t + \beta(T-t)\nabla \log q(R_t, T-t)\right]dt + \sqrt{\beta(T-t)}dW_t, \tag{11}$$

With $R_0 \sim \rho(r)$ as initial condition and time $t$ flows from $t = 0$ to $t = T$.

Once the parametric score network is optimized, trough the simulation of Equation (11), sampling $R_T \sim q_\psi(r)$ is possible allowing **joint generation**. A numerical SDE solver can be used to sample $R_T$ which can be fed to the modality specific decoders to jointly sample a set of $\hat{X} = \{d_\theta^i(R_T^i)\}_{i=0}^M$. As explained in Section 4.2, the use of the unconditional score network $s_\chi(R_t, t)$ allows **conditional generation** through the approximation described in Song et al. (2021b).

As described in Algorithm 1, one can generate a set of modalities $A_1$ conditioned on the available set of modalities $A_2$. First, the available modalities are encoded into their respective latent space $z^{A_2}$, the initial missing part is sampled from the stationary distribution $R_0^{A_1} \sim \rho(r^{A_1})$, using an SDE solver (e.g. Euler-Maruyama), the reverse diffusion SDE (in Equation (11)) is discretized using a finite time steps $\Delta t = T/N$, starting from $t = 0$ and iterating until $t \approx T$. At each iteration, the available portion of the latent space is diffused and brought to the same noise level as $R_t^{A_1}$ allowing the use of the unconditional score network. Lastly, the reverse diffusion update is done. This process is repeated until arriving at $t \approx T$ and obtaining $R_T^{A_1} = \hat{Z}^{A_1}$ which can be decoded to recover $\hat{x}^{A_1}$. Note that the joint generation can be seen as a special case of Algorithm 1 with $A_2 = \emptyset$. We name this first approach Multi-modal Latent Diffusion with In-painting (MLD IN-PAINT) and provide extensive comparison with our method MLD in Appendix B.

---

**Algorithm 1:** MLD IN-PAINT conditional generation

**Data:** $x^{A_2} = \{x^i\}_{i \in A_2}$
$z^{A_2} \leftarrow \{e_{\phi_i}(x^i)\}_{i \in A_2}$ // Encode the available modalities $X$ into their latent space
$A_1 \leftarrow \{1, \dots, M\} \setminus A_2$ // The set of modalities to generate
$R_0 \leftarrow \mathcal{C}(R_0^{A_1}, z^{A_2}), \qquad R_0^{A_1} \sim \rho(r^{A_1})$ // Compose the initial state
$R \leftarrow R_0$
$\Delta t \leftarrow T/N$
**for** $n = 0$ **to** $N - 1$ **do**
    $t' \leftarrow T - n\,\Delta t$
    $\bar{R} \sim q(r|R_0, t')$ // Diffuse the available portion of the latent space (eq. (10))
    $R \leftarrow m(A_1) \odot R + (1 - m(A_1)) \odot \bar{R}$
    $\epsilon \sim \mathcal{N}(0; I)$ **if** $n < (N-1)$ **else** $\epsilon = 0$
    $\Delta R \leftarrow \Delta t \left[\frac{1}{2}\beta(t')R + \beta(t')s_\chi(R, t')\right] + \sqrt{\beta(t')\Delta t}\epsilon$
    $R \leftarrow R + \Delta R$ // The Euler-Maruyama update step
**end**
$\hat{z}^{A_1} \leftarrow R^{A_1}$
**Return** $\hat{X}^{A_1} = \{d_\theta^i(\hat{z}^i)\}_{i \in A_1}$

---

As discussed in Section 4.2, the approximation enabling the in-painting approach can be efficient in several domains but its generalization to the multi-modal latent space scenario is not trivial. We argue that this is due to the heterogeneity of modalities which induce different latent spaces characteristics. For different modality-specific latent spaces, the loss of information ratio can vary through the diffusion process. We verify this hypothesis through the following experiment.

**Latent space robustness against diffusion perturbation:**    We analyse the effect of the forward diffusion perturbation on the latent space through time. We encode the modalities using their respective encoders to obtain their latent space $Z = [e_{\psi^1}(X^1) \dots e_{\psi^M}(X^M)]$. Given a time $t \in [0, T]$, we diffuse the different latent spaces by applying Equation (10) to get $R_t \sim q(r|z, t)$ with $R_t$ being the perturbed version of the latent space at time $t$. We feed the modality specific decoders with the perturbed latent space $\hat{X}_t = \{d_\theta^i(R_t^i)\}_{i=1}^M$, $\hat{X}_t$ being the output modalities generated using the perturbed latent space. To evaluate the information loss induced by the diffusion process on the different modalities, we assess the coherence preservation in the reconstructed modalities $\hat{X}_t$ by computing the coherence (in %) as done in Section 5.

We expect to obtain high coherence results for $t \approx 0$, when compared to $t \approx T$, the information in the latent space being more preserved at the beginning of the diffusion process than at the last phase of the froward SDE where all dependencies on initial conditions vanish. Figure 5 shows the coherence as a function of the diffusion time $t \in [0, 1]$ for different modalities across multiple data-sets. We observe that within the same data-set, some modalities stand out with a specific level of robustness (using as a proxy the coherence level) against the diffusion perturbation in comparison with the remaining modalities from the same data-set. For instance, we remark that SVHN is less robust than MNIST which should manifest in an under-performance of SVHN to MNIST conditional generation. An intuition that we verify in Appendix B.

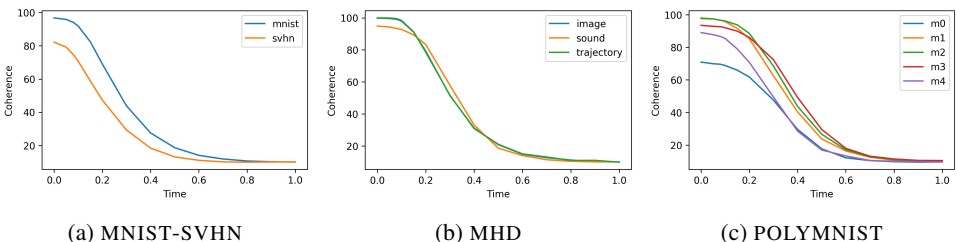

(a) MNIST-SVHN                (b) MHD                (c) POLYMNIST

Figure 5: The coherence as a function of the diffusion process time for three datasets. The diffusion perturbation is applied on the modalities latent space after an element wise normalization.

### A.3    MULTI-TIME MASKED MULTI-MODAL SDE

To learn the score network capable of both conditional and joint generation, we proposed in Section 4 a multi-time masked diffusion process.

Algorithm 2 presents a pseudo-code for the multi time masked training. The masked diffusion process is applied following a randomization with probably $d$. First, a subset of modalities $A_2$ is selected randomly to be the conditioning modalities and $A_1$ the remaining set of modalities to be the diffused modalities. The time $t$ is sampled uniformly from $[0, T]$ and the portion of the latent space corresponding to the subset $A_1$ is diffused accordingly. Using the masking as shown in Algorithm 2, the portion of the latent space corresponding to the subset $A_2$ is not diffused and forced to be equal to $R_0^{A_2} = z^{A_2}$. The multi-time vector $\tau$ is constructed. Lastly, the score network is optimized by minimizing a masked loss corresponding to the diffused part of the latent space. With probability $(1 - d)$, all the modalities are diffused at the same time and $A_2 = \emptyset$. In order to calibrate the loss, given that the randomization of $A_1$ and $A_2$ can result in diffusing different sizes of the latent space, we re-weight the loss according to the cardinality of the diffused and freezed portions of the latent space:

$$\Omega(A_1, A_2) = 1 + \frac{dim(A_2)}{dim(A_1)} \tag{12}$$

Where $\dim(.)$ is the sum of each latent space cardinality of a given subset of modalities with $dim(\emptyset) = 0$.

---

**Algorithm 2:** MLD Masked Multi-time diffusion training step

---

**Data:** $X = \{x^i\}_{i=1}^M$
**Param:** $d$
$Z \leftarrow \{e_{\phi_i}(x^i)\}_{i=0}^M$       // Encode the modalities $X$ into their latent space
$A_2 \sim \nu$                   // $\nu$ depends on the parameter $d$
$A_1 \leftarrow \{1, \ldots, M\} \setminus A_2$
$t \sim \mathcal{U}[0, T]$
$R \sim q(r|Z, t)$       // Diffuse the available portion of the latent space (Equation (10))
$R \leftarrow m(A_1) \odot R + (1 - m(A_1)) \odot Z$       // Masked diffusion
$\tau(A_1, t) \leftarrow [\mathbb{1}(1 \in A_1)t, \ldots, \mathbb{1}(M \in A_1)t]$     // Construct the multi time vector
**Return** $\nabla_\chi \left\{ \Omega(A_1, A_2) \quad \left\| m(A_1) \odot \quad [s_\chi(R, \tau(A_1, t)) - \nabla \log q(R, t|z^{A_2})] \right\|_2^2 \right\}$

---

The optimized score network can approximate both the conditional and unconditional true score:
$$s_\chi(R_t, \tau(A_1, t)) \sim \nabla \log q(R_t, t \mid z^{A_2})). \tag{13}$$
The joint generation is a special case of the latter with $A_2 = \emptyset$:
$$s_\chi(R_t, \tau(A_1, t)) \sim \nabla \log q(R_t, t) \quad , A_1 = \{1, ..., M\} \tag{14}$$

Algorithm 3 describes the reverse conditional generation pseudo-code. It's pertinent to compare this algorithm with Algorithm 1. The main difference resides in the use of the multi-time score network, enabling conditional generation with the multi-time vector playing the role of time information and conditioning signal. On the other hand, in Algorithm 1, we don't have a conditional score network, therefore we resort to the approximation from Section 4.2, and use the unconditional score.

---

**Algorithm 3:** MLD conditional generation.

---

**Data:** $x^{A_2} \leftarrow \{x^i\}_{i \in A_2}$
$z^{A_2} \leftarrow \{e_{\phi_i}(x^i)\}_{i \in A_2}$   // Encode the available modalities $X$ into their latent space
$A_1 \leftarrow \{1, \ldots, M\} \setminus A_2$       // The set of modalities to be generated
$R_0 \leftarrow \mathcal{C}(R_0^{A_1}, z^{A_2}), \quad R_0^{A_1} \sim \rho(r^{A_1})$   // Compose the initial latent space
$R \leftarrow R_0$
$\Delta t \leftarrow T/N$
**for** $n = 0$ **to** $N - 1$ **do**
    $t' \leftarrow T - n\,\Delta t$
    $\tau(A_1, t') \leftarrow [\mathbb{1}(1 \in A_1)t', \ldots, \mathbb{1}(M \in A_1)t']$     // Construct the multi-time vector
    $\epsilon \sim \mathcal{N}(0; I)$   **if** $n < N$   **else**   $\epsilon = 0$
    $\Delta R \leftarrow \Delta t \left[\frac{1}{2}\beta(t')R + \beta(t')s_\chi(R, \tau(A_1, t'))\right] + \sqrt{\beta(t')\Delta t}\epsilon$
    $R \leftarrow R + \Delta R$           // The Euler–Maruyama update step
    $R \leftarrow m(A_1) \odot R + (1 - m(A_1)) \odot R_0$       // Update the portion corresponding to the unavailable modalities
**end**
$\hat{z}^{A_1} = R^{A_1}$
**Return** $\hat{X}^{A_1} = \{d_\theta^i(\hat{z}^i)\}_{i \in A_1}$

---

### A.4 UNI-DIFFUSER TRAINING

The work presented in Bao et al. (2023) is specialized for an image-caption application. The approach is based on a multi-modal diffusion model applied to a unified latent embedding, obtained via

pre-trained autoencoders, and incorporating pre-trained models (CLIP Radford et al. (2021) and GPT-2 Radford et al. (2019)). The unified latent space is composed of an image embedding, a CLIP image embedding and a CLIP text embedding. Note that the CLIP model is pre-trained on a pairs of multi-modal data (image-text), which is expected to enhance the generative performance. Since it is not trivial to have a jointly trained encoder similar to CLIP for any type of modality, the evaluation of this model on different modalities across different data-set (e.g. including audio) is not an easy task.

To compare to this work, we adapt the training scheme presented in Bao et al. (2023) to our MLD method. Instead of applying a masked multi-modal SDE for training the score network, every portion of the latent space is diffused according to a different time $t^i \sim \mathcal{U}(0, 1)$ and, therefore, the multi-time vector fed to the score network is $\tau(t) = [t^0 \sim \mathcal{U}(0, 1), ..., t^M \sim \mathcal{U}(0, 1)]$. For fairness, we use the same score network and reverse process sampler as for our MLD version with multi-time training, and call this variant Multi-modal Latent Diffusion UniDiffuser (MLD UNI).

### A.5 Intuitive summary: How does MLD capture modality interactions?

MLD treats the latent spaces of each modality as variables that evolve differently through the diffusion process according to a multi-time vector. The masked multi-time training enables the model to learn the score of all the combination of conditionally diffused modalities, using the frozen modalities as the conditioning signal, through a randomized scheme. By learning the score function of the diffused modalities at different time steps, the score model captures the correlation between the modalities. At test time, the diffusion time of each modality is chosen to modulate its influence on the generation, as follows.

For joint generation the model uses the unconditional score which corresponds to using the same diffusion time for all modalities. Thus, all the modalities influence each other equally. This ensures that modality interaction information is faithful to the one characterizing the observed data distribution.

The model can also generate modalities conditionally by using the conditional score, by freezing the conditioning modalities during the reverse process. The freezed state is similar to the final state of the revere process where information is not perturbed, thus the influence of the conditioning modalities is maximal. Subsequently, the generated modalities reflect the necessary information from the conditioning modalities and achieve the desired correlation.

### A.6 Technical details

**Sampling schedule:**   We use the sampling schedule proposed in Lugmayr et al. (2022), which has shown to improve the coherence of the conditional and joint generation. We use the best parameters suggested by the authors: $N = 250$ time-steps, applied $r = 10$ re-sampling times with jump size $j = 10$. For readability in algorithm 1 and algorithm 3, we present pseudo code with a linear sampling schedule which can be easily adapted to any other schedule.

**Training the score network:**   Inspired by the architecture from (Dupont et al., 2022), we use simple Residual MLP blocks with skip connections as our score network (see Figure 6). We fix the **width** and **number of blocks** proportionally to the number of the modalities and the latent space size. As in Song & Ermon (2020), we use Exponential moving average (EMA) of model parameters with a momentum parameter $m = 0.999$.

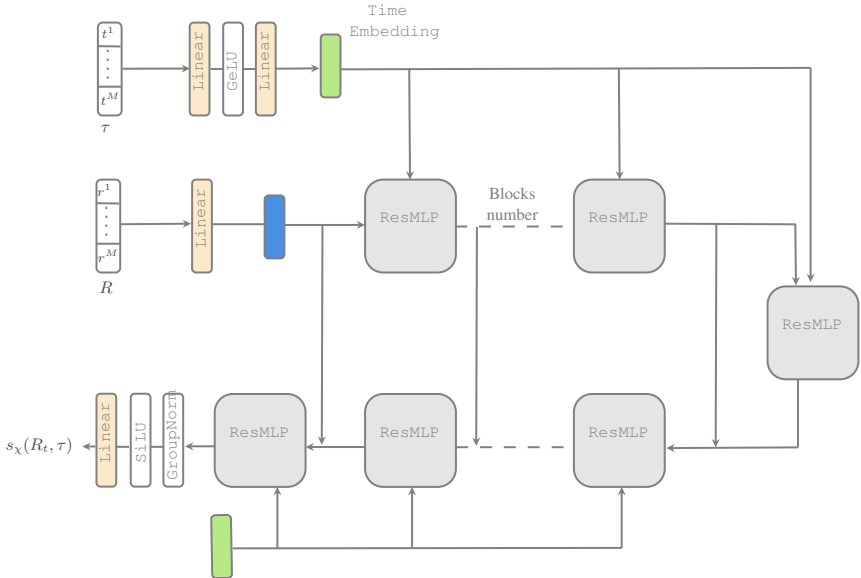

Figure 6: Score network $s_\chi$ architecture used in our MLD implementation. Residual MLP block architecture is shown in Figure 7.

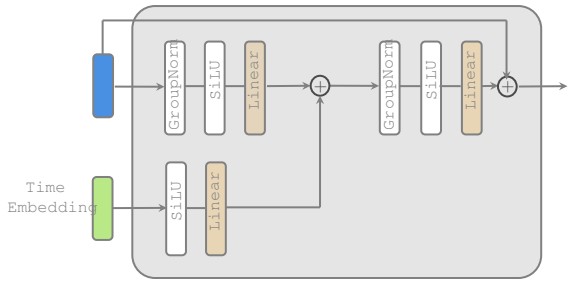

Figure 7: Architecture of ResMLP block.

# B    MLD ABLATIONS STUDY

In this section, we compare MLD with two variants presented in Appendix A : MLD IN-PAINT, a naive approach without our proposed *multi-time masked* SDE, MLD UNI a variant of our method using the same training scheme of Bao et al. (2023). We also analyse the effect of the $d$ randomization parameter on MLD performance through ablations study.

## B.1    MLD AND ITS VARIANTS

Table 4 summarizes the different approaches adopted in each variant. All the considered models share the same deterministic autoencoders trained during the first stage.

For fairness, our evaluation was done using the same configuration and code basis of MLD. This includes: the autoencoder architectures and latent space size (similar to Section 5), the same score network (Figure 6) is used across experiments, with MLD IN-PAINT using the same architecture with one time dimension instead of the multi-time vector. In all the variants, the joint and conditional generation are conducted using the same reverse sampling schedule described in Appendix A.6.

Table 4: MLD and its variants ablation study

| Model | Multi-time diffusion | Training | Conditional and joint generation |
|---|---|---|---|
| MLD IN-PAINT | x | Equation (5) | Algorithm 1 |
| MLD UNI | ✓ | Bao et al. (2023) | Algorithm 3 |
| MLD | ✓ | Algorithm 2 | Algorithm 3 |

**Results** In some cases, the MLD variants can match the joint generation performance of MLD but, overall, they are less efficient and have noticeable weaknesses: MLD IN-PAINT under-performs in conditional generation when considering relatively complex modalities, MLD UNI is not able to leverage the presence of multiple modalities to improve cross generation, especially for data-sets with a large number of modalities. On the other hand, MLD is able to overcome all these limitations.

**MNIST-SVHN.** In Table 5, MLD achieves the best results and dominates cross generation performance. We observe that MLD IN-PAINT lacks coherence for SVHN to MNIST conditional generation, a results we expected by analysing the experiment in Figure 5. MLD UNI, despite the use of a multi-time diffusion process, under-performs our method, which indicates the effectiveness of our masked diffusion process in learning the conditional score network. Since all the models use the same deterministic autoencoders, the observed generative quality performance are relatively similar (See Figure 8 for qualitative results ).

Table 5: Generation Coherence and Quality for MNIST-SVHN (M is for MNIST and S for SVHN ). The generation quality is measured in terms of FMD for MNIST and FID for SVHN.

| Models | Coherence (%↑) | | | Quality (↓) | | | |
|---|---|---|---|---|---|---|---|
| | Joint | M → S | S → M | Joint(M) | Joint(S) | M → S | S → M |
| MLD-Inpaint | $85.53_{\pm 0.22}$ | $\underline{81.76}_{\pm 0.23}$ | $63.28_{\pm 1.16}$ | $\mathbf{3.85}_{\pm 0.02}$ | $60.86_{\pm 1.27}$ | $59.86_{\pm 1.18}$ | $\mathbf{3.55}_{\pm 0.11}$ |
| MLD-Uni | $82.19_{\pm 0.97}$ | $79.31_{\pm 1.21}$ | $\underline{72.78}_{\pm 1.81}$ | $4.1_{\pm 0.17}$ | $57.41_{\pm 1.43}$ | $\underline{57.84}_{\pm 1.57}$ | $4.84_{\pm 0.28}$ |
| MLD | $\underline{85.22}_{\pm 0.5}$ | $\mathbf{83.79}_{\pm 0.62}$ | $\mathbf{79.13}_{\pm 0.38}$ | $\underline{3.93}_{\pm 0.12}$ | $\mathbf{56.36}_{\pm 1.63}$ | $\mathbf{57.2}_{\pm 1.47}$ | $\underline{3.67}_{\pm 0.14}$ |

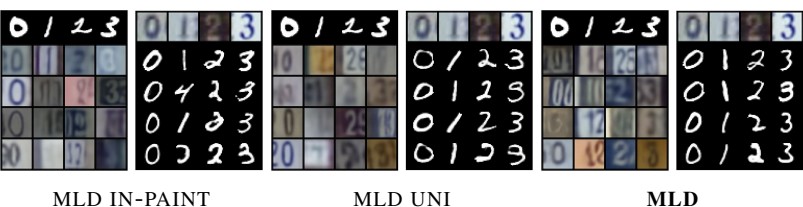

MLD IN-PAINT                 MLD UNI                 **MLD**

Figure 8: Qualitative results for **MNIST-SVHN**. For each model we report: MNIST to SVHN conditional generation in the left, SVHN to MNIST conditional generation in the right.

**MHD.** Table 6 shows the performance results for the MHD data-set in terms of generative coherence. MLD achieves the best joint generation coherence and, along with MLD UNI, they dominate the cross generation coherence. MLD IN-PAINT shows a lack of coherence when conditioning on the sound modality alone, a predictable result since this is a more difficult configuration, as the sound modality is loosely correlated to other modalities. We also observe that MLD IN-PAINT performs worse than the two other alternatives when conditioned on the trajectory modality, which is the smallest modality in terms of latent size. This indicates another limitation of the naive approach regarding coherent generation when handling different latent spaces sizes, a weakness our method MLD overcomes. Table 7 presents the qualitative generative performance which are homogeneous across the variants with MLD, achieving either the best or second best performance.

Table 6: Generation Coherence (%↑) for MHD (Higher is better). Line above refers to the generated modality while the observed modalities subset are presented below.

| Models | Joint | I (Image) | | | T (Trajectory) | | | S (Sound) | | |
|---|---|---|---|---|---|---|---|---|---|---|
| | | T | S | T,S | I | S | I,S | I | T | I,T |
| MLD-Inpaint | $96.88_{\pm0.35}$ | $63.9_{\pm1.7}$ | $56.52_{\pm1.89}$ | $95.83_{\pm0.48}$ | $\underline{99.58}_{\pm0.1}$ | $56.51_{\pm1.89}$ | $\underline{99.89}_{\pm0.04}$ | $95.81_{\pm0.25}$ | $56.51_{\pm1.89}$ | $96.38_{\pm0.35}$ |
| MLD-Uni | $\underline{97.69}_{\pm0.26}$ | $\mathbf{99.91}_{\pm0.04}$ | $\mathbf{89.87}_{\pm0.38}$ | $\mathbf{99.92}_{\pm0.04}$ | $\mathbf{99.68}_{\pm0.1}$ | $\mathbf{89.78}_{\pm0.45}$ | $99.38_{\pm0.31}$ | $\underline{97.54}_{\pm0.2}$ | $\underline{97.65}_{\pm0.41}$ | $\underline{97.79}_{\pm0.41}$ |
| MLD | $\mathbf{98.34}_{\pm0.22}$ | $99.45_{\pm0.09}$ | $\underline{88.91}_{\pm0.54}$ | $\underline{99.88}_{\pm0.04}$ | $\underline{99.58}_{\pm0.03}$ | $\underline{88.92}_{\pm0.53}$ | $\mathbf{99.91}_{\pm0.02}$ | $\mathbf{97.63}_{\pm0.14}$ | $\mathbf{97.7}_{\pm0.34}$ | $\mathbf{98.01}_{\pm0.21}$ |

Table 7: Generation quality for MHD. The metrics reported are FMD for Image and Trajectory modalities and FAD for the sound modalities (Lower is better).

| Models | I (Image) | | | | T (Trajectory) | | | | S (Sound) | | | |
|---|---|---|---|---|---|---|---|---|---|---|---|---|
| | Joint | T | S | T,S | Joint | I | S | I,S | Joint | I | T | I,T |
| MLD-Inpaint | $5.35_{\pm1.35}$ | $6.23_{\pm1.13}$ | $\underline{4.76}_{\pm0.68}$ | $3.53_{\pm0.36}$ | $\mathbf{1.59}_{\pm0.12}$ | $\mathbf{0.6}_{\pm0.05}$ | $\mathbf{1.81}_{\pm0.13}$ | $\mathbf{0.54}_{\pm0.06}$ | $2.41_{\pm0.07}$ | $2.5_{\pm0.04}$ | $2.52_{\pm0.02}$ | $2.49_{\pm0.05}$ |
| MLD-Uni | $7.91_{\pm2.2}$ | $\mathbf{1.65}_{\pm0.33}$ | $6.29_{\pm1.38}$ | $\underline{3.06}_{\pm0.54}$ | $\underline{2.53}_{\pm0.5}$ | $1.18_{\pm0.26}$ | $3.18_{\pm0.77}$ | $2.84_{\pm1.14}$ | $\mathbf{2.11}_{\pm0.08}$ | $\mathbf{2.25}_{\pm0.05}$ | $\mathbf{2.1}_{\pm0.0}$ | $\mathbf{2.15}_{\pm0.01}$ |
| MLD | $\underline{7.98}_{\pm1.41}$ | $\underline{1.7}_{\pm0.14}$ | $\mathbf{4.54}_{\pm0.45}$ | $\mathbf{1.84}_{\pm0.27}$ | $3.18_{\pm0.18}$ | $\underline{0.83}_{\pm0.03}$ | $\underline{2.07}_{\pm0.26}$ | $\underline{0.6}_{\pm0.05}$ | $\underline{2.39}_{\pm0.1}$ | $\underline{2.31}_{\pm0.07}$ | $\underline{2.33}_{\pm0.11}$ | $\underline{2.29}_{\pm0.06}$ |

**POLYMNIST.** In Figure 9, we remark the superiority of MLD in both generative coherence and quality. MLD-Uni is not able to leverage the presence of a large number of modalities in conditional generation coherence. Interestingly, an increase in the number of input modalities impacts negatively the performance of MLD UNI.

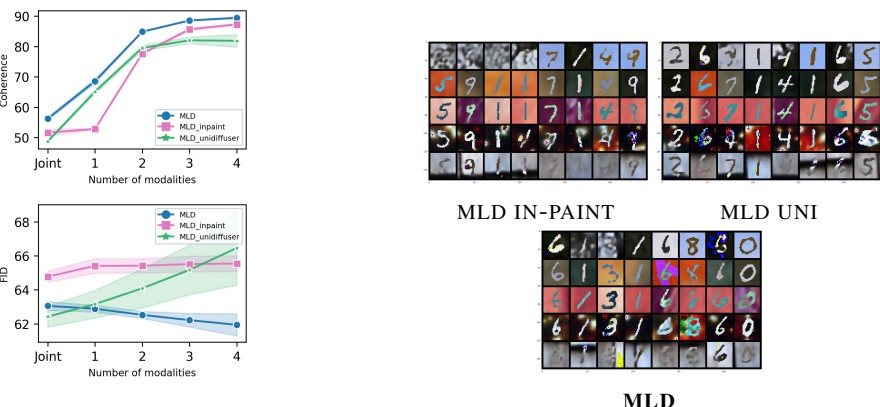

Figure 9: Results for **POLYMNIST** data-set. *Left*: a comparison of the generative coherence (% ↑) and quality in terms of FID (↓)) as a function of the number of modality input. We report the average performance following the leave-one-out strategy (see Appendix C). *Right*: are qualitative results for the joint generation of the 5 modalities.

**CUB.** Figure 10 shows qualitative results for caption to image conditional generation. All the variants are based on the same first stage autoencoders, and the generative performance in terms of quality are comparable.

### B.2 RANDOMIZATION $d$-ABLATIONS STUDY

The $d$ parameter controls the randomization of the *multi-time masked diffusion process* during training in Algorithm 2. With probability $d$, the concatenated latent space corresponding to all the modalities is diffused at the same time. With probability $(1 - d)$, a portion of the latent space corresponding to a random subset of the modalities is not diffused and freezed during the training step. To study the parameter $d$ and its effect on the performance of our MLD model, we use $d \in \{0.1, .., 0.9\}$. Figure 11 shows the $d$-ablations study results on the **MNIST-SVHN** dataset. We report the performance results averaged over 5 independent seeds as a function of the probability $(1 - d)$ : **Left:** the conditional

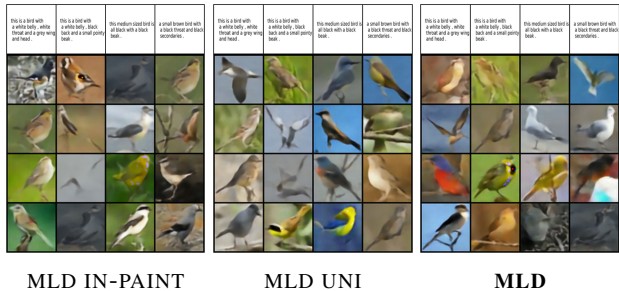

MLD IN-PAINT      MLD UNI      **MLD**

Figure 10: Qualitative results on **CUB** data-set. Caption used as condition to generate the bird images.

and joint coherence for **MNIST-SVHN** dataset. **Middle:** the quality performance in terms of FID for SVHN generation. **Right:** the quality performance in terms of FMD for MNIST generation.

We observe that higher value for $1 - d$ thus greater probability of applying the *multi-time masked diffusion*, improves the SVHN to MNIST conditional generation coherence. This confirms that the masked multi-time training enables better conditional generation. Overall, on the **MNIST-SVHN** dataset, MLD shows weak sensibility to the $d$ parameter whenever the value of $d \in [0.2, 0.7]$.

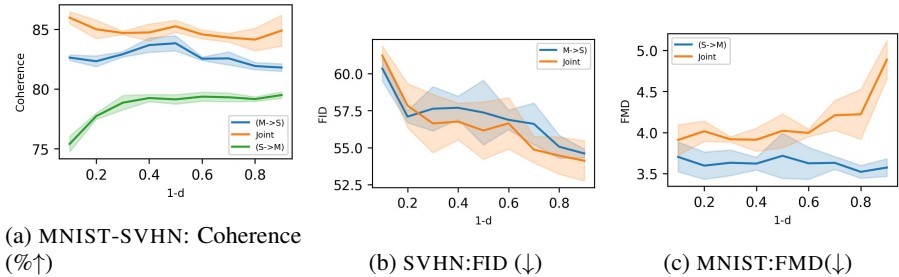

(a) MNIST-SVHN: Coherence (%↑)

(b) SVHN:FID (↓)

(c) MNIST:FMD(↓)

Figure 11: The randomization parameter $d$ ablations study on **MNIST-SVHN**.

## C DATASETS AND EVALUATION PROTOCOL

### C.1 DATASETS DESCRIPTION

**MNIST-SVHN** Shi et al. (2019) is constructed using pairs of MNIST and SVHN, sharing the same digit class (See Figure 12a). Each instance of a digit class (in either dataset) is randomly paired with 20 instances of the same digit class from the other data-set. SVHN modality samples are obtained from house numbers in Google Street View images, characterized by a variety of colors, shapes and angles. A high number of SVHN samples are noisy and can contain different digits within the same sample due to the imperfect cropping of the original full house number image. One challenge of this data-set for multi-modal generative models is to learn to extract digit number and reconstruct a coherent MNIST modality.

**MHD** Vasco et al. (2022) is composed of 3 modalities: synthetically generated images and motion trajectories of handwritten digits associated with their speech sounds. The images are gray scale $1 \times 28 \times 28$ and the handwriting trajectory are represented by a $1 \times 200$ vector. The spoken digits sound is $1s$ long audio processed as Mel-Spectrograms constructed with a hopping window of $512$ ms with 128 Mel Bins resulting in a $1 \times 128 \times 32$ representation. This benchmark is the closest to a real world multi-modal sensors scenario because of the presence of three completely different modalities, the audio modality representing a complex data type. Therefore, similar to SVHN, the conditional generation of sound to coherent images or trajectories represents a challenging use case.

**POLYMNIST** Sutter et al. (2021) is an extended version of the MNIST data-set to 5 modalities. Each modality is constructed using a randomly set of MNIST digits with an overlay over a random crop from a modality specific, 3 channel image background. This synthetic generated data-set allows the evaluation of the scalability of multi-modal generative models to large number of modalities. Although this data-set is composed of only images, the different modality-specific background having different textures, results in different levels of difficulty. In Figure 12c, the digits numbers are more difficult to distinguish in modality 1 and 5 than in the remaining modalities.

**CUB** Shi et al. (2019) is comprised of bird images and their associated text captions. The work in Shi et al. (2019) used a simplified version based on pre-computed ResNet-features. We follow Daunhawer et al. (2022) and conduct all our experiments on the real image data instead. Each image from the 11,788 photos of birds from Caltech-Birds Wah et al. (2011) are resized to $3 \times 64 \times 64$ image size and coupled with 10 textual descriptions of the respective bird (See Figure 12d).

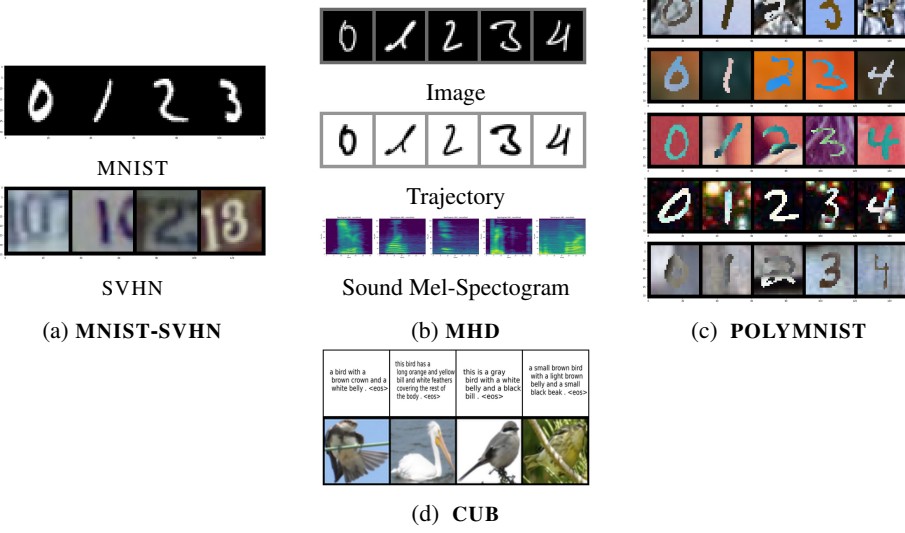

Figure 12: Illustrative example of the Datasets used for the evaluation

### C.2 EVALUATION METRICS

Multimodal generative models are evaluated in terms of generative coherence and quality.

#### C.2.1 GENERATION COHERENCE

We measure *coherence* by verifying that generated data (both for joint and conditional generations) share the same information across modalities. Following Shi et al. (2019); Sutter et al. (2021); Hwang et al. (2021); Vasco et al. (2022); Daunhawer et al. (2022), we consider the class label of the modalities as the shared information and use pre-trained classifiers to extract the label information form the generated samples and compare it across modalities.

For **MNIST-SVHN** , **MHD** and **POLYMNIST**, the shared semantic information is the digit class number. Single modality classifiers are trained to classify the digit number of a given modality sample. To compute the conditional generation of modality $m$ with a subset of modalities $A$, we feed the modality specific pre-trained classifier $\mathbf{C}_m$ with the conditional generated sample $\hat{X}^m$. The predicted label class is compared to the ground truth label $y_{X^A}$ which is the label of modalities of the subset $X^A$. For $N$ samples, the matching rate average establishes the coherence. For all the experiments, $N$ is equal to the length of the test-set.

$$Coherence(\hat{X}^m|X^A) = \frac{1}{N} \sum_{1}^{N} \mathbb{1}_{\{\mathbf{C}_m(\hat{X}^m)=y_{X^A}\}} \tag{15}$$

The **joint generation coherence** is measured by feeding the generated samples of each modality to their specific trained classifier. The rate with which all classifiers output the same predicted digit label for $N$ generations is considered as the joint generation coherence.

**The leave one out coherence:** is the conditional generation coherence using all the possible subsets excluding the generated modality: $Coherence(\hat{X}^m|X^A)$ with $A = \{1,..,M\} \setminus m$ ). Due to the large number of modalities in **POLYMNIST**, similar to Sutter et al. (2021); Hwang et al. (2021); Daunhawer et al. (2022) we compute the average **leave one out coherence** conditional coherence as a function of the input modalities subset size.

Due to the unavailability of labels in the **CUB** data-set, we use CLIP-S Hessel et al. (2021) a state of the art metric for image captioning evaluation.

#### C.2.2 GENERATION QUALITY

For each modality, we consider the following metrics:

- **RGB Images**: FID Heusel et al. (2017) is the state-of-the-art standard metric to evaluate image generation quality of generative models.
- **Audio**: FAD Kilgour et al. (2019), is state-of-the-art standard metric in the evaluation of audio generation. FAD performs well in terms of robustness against noise and is consistent with human judgments Vinay & Lerch (2022). Similar to FID, a Fréchet distance is computed but VGGish (audio classifer model) embeddings are used instead.
- **Other modalities** For other modality types, we derive FMD (Fréchet Modality Distance), a similar metric to FID and FAD. We compute the **Fréchet distance** between the statistics retrieved from the activations of the modality specific pre-trained classifiers used for coherence evaluation. FMD is used to evaluate the generative quality of MNIST modality in **MNIST-SVHN** and image and trajectory modalities in **MHD** data-set.

For conditional generation, we compute the quality metric (FID,FAD or FMD) using the conditionally generated modality and the real data. For joint generation, we use the randomly generated modality and randomly selected same number of samples from the real data.

For **CUB**, we use 10000 samples to evaluate the generation quality in terms of FID. In the remaining experiments, we use 5000 samples to evaluate the performance in terms of FID, FAD or FMD.

# D  IMPLEMENTATION DETAILS

We report in this section the implementation details for each benchmark. We used the same unified code-base for all the baselines, using the *PyTorch* framework. The VAE implementation is adapted from the official code whenever it's available (MVAE, MMVAE and MOPOE as in [3], MVTCAE [4] and NEXUS[5] ). For fairness, MLD and all the VAE-based models use the same autoencoder architecture. We use the best hyper-parameters suggested by the authors. Across all the data-sets, we use the *Adam optimizer* Kingma & Ba (2014) for training.

## D.1  MLD

MLD uses the same autoencoders architecture used for VAE-based models, except that these are deterministic autoencoders. The autoencoders are trained using the same reconstruction loss term as for the VAE-based models. Table 8 and Table 9 summarize the hyper-parameters used during the two phases of MLD training. Note that for the image modality in the CUB dataset, to overcome over-fitting in training the deterministic autoencoder, data augmentation was necessary (we used *TrivialAugmentWide* from the Torchvision library).

Table 8: MLD: The deterministic autoencoders hyper-parameters

| Dataset | Modality | Latent space | Batch size | Lr | Epochs | Weight decay |
|---------|----------|--------------|------------|-----|--------|--------------|
| **MNIST-SVHN** | MNIST
SVHN | 16
64 | 128 | 1e-3 | 150 | |
| **MHD** | Image
Trajectory
Sound | 64
16
128 | 64 | 1e-3 | 500 | |
| **POLYMNIST** | All modalities | 160 | 128 | 1e-3 | 300 | |
| **CUB** | Caption
Image | 32
64 | 128 | 1e-3
1e-4 | 500
300 | 1e-6 |
| **CelebAMask-HQ** | Image
Mask
Attributes | 256
128
32 | 64 | 1e-3 | 200 | |

Table 9: MLD: The score network hyper-parameters

| Dataset | $d$ | Blocks | Width | Time embed | Batch size | Lr | Epochs |
|---------|-----|--------|-------|------------|------------|-----|--------|
| **MNIST-SVHN** | 0.5 | 2 | 512 | 256 | 128 | | 150 |
| **MHD** | 0.3 | 2 | 1024 | 512 | 128 | 1e-4 | 3000 |
| **POLYMNIST** | 0.5 | 2 | 1536 | 512 | 256 | | 3000 |
| **CUB** | 0.7 | 2 | 1024 | 512 | 64 | | 3000 |
| **CelebAMask-HQ** | 0.5 | 2 | 1536 | 512 | 64 | | 3000 |

## D.2  VAE-BASED MODELS

For **MNIST-SVHN**, we follow Sutter et al. (2021); Shi et al. (2019) and use the same autoencoder architecture and pre-trained classifier.The latent space size is set to 20, $\beta = 5.0$. For MVTCAE $\alpha = \frac{5}{6}$. For both modalities, the likelihood is estimated using Laplace distribution. For NEXUS, we use the same modalities latent space sizes as in MLD, the joint NEXUS latent space is set to 20, $\beta_i = 1.0$ and $\beta_c = 5.0$. We train all the VAE-models for 150 epochs with 256 batch size and learning rate of $1e - 3$.

---

[3]https://github.com/thomassutter/MoPoE
[4]https://github.com/gr8joo/MVTCAE
[5]https://github.com/miguelsvasco/nexus_pytorch

For **MHD**, we reuse the autoencoders architecture and pre-trained classifier of Vasco et al. (2022). We adopt the hyper-parameters of Vasco et al. (2022) to train NEXUS model with the same settings, besides discarding the label modality. For the remaining VAE-based models, the latent space size is set to 128, $\beta = 1.0$ and $\alpha = \frac{5}{6}$ for MVTCAE. For all the modalities, Mean square error (MSE) is used to compute the reconstruction loss, similar to Vasco et al. (2022). These models are trained for 600 epochs with 128 batch size and learning rate of $1e - 3$.

For **POLYMNIST**, we use the same autoencoders architecture and pretrained classifier used by Sutter et al. (2021); Hwang et al. (2021). We set the latent space size to 512, $\beta = 2.5$ and $\alpha = \frac{5}{6}$ for MVTCAE. For all the modalities, the likelihood is estimated using Laplace distribution. For NEXUS, we use the same modality latent space size as in MLD, the joint NEXUS latent space to 64, $\beta_i = 1.0$ and $\beta_c = 2.5$. We train all the models for 300 epochs with 256 batch size and learning rate of $1e - 3$.

For **CUB**, we use the same autoencoders architecture and implementation settings as in Daunhawer et al. (2022). Laplace and one-hot categorical distributions are used to estimate likelihoods of the image and caption modalities respectively. The latent space size is set to 64, $\beta = 9.0$ for MVAE, MVTCAE and MOPOE and $\beta = 1$ for MMVAE. We set $\alpha = \frac{5}{6}$ for MVTCAE. For NEXUS, we use the same modalities latent space sizes as in MLD, the joint NEXUS latent space is set to 64, $\beta_i = 1.0$ and $\beta_c = 1$. We train all the models for 150 epochs with 64 batch size, with learning rate of $5e - 4$ for MVAE, MVTCAE and MOPOE and $1e - 3$ for the remaining models.

Finally, note that in the official implementation of Sutter et al. (2021) and Hwang et al. (2021), for the **POLYMNIST** and **MNIST-SVHN** data-sets, the classifiers were used for evaluation using dropout. In our implementation, we make sure to deactivate dropout during evaluation step.

### D.3 MLD WITH POWERFULL AUTOENCODER

Here we provide more detail about the CUB experiment using more powerful autoencoder denoted MLD* in Figure 3. We use an architecture similar to Rombach et al. (2022) adapted to (64X64) resolution images. We modified the autoencoder architecture to be deterministic and train the model with a simple Mean square error loss. We kept the same configuration of the CUB experiment described in the previous experiment on the same dataset including the text autoencoder, score network and hyper-parameters. We also perform experiments with the same settings on (128X128) resolution images. We included the qualitative results in fig. 25.

### D.4 COMPUTATION RESOURCES

In our experiments, we used 4 A100 GPUs, for a total of roughly 4 months of experiments.

# E ADDITIONAL RESULTS

In this section, we report detailed results for all of our experiments, including standard deviation and additional qualitative samples, for all the data-sets and all the methods we compared in our work.

## E.1 MNIST-SVHN

### E.1.1 SELF RECONSTRUCTION

In Table 10 we report results about *self-coherence*, which we use to support the arguments from Section 2. This metric is used to measure the loss of information due to latent collapse, by showing the ability of all competing models to reconstruct an arbitrary modality given the same modality or a set thereof as an input. For our MLD model, the self-reconstruction is done without using the diffusion model component: the modality is encoded using its deterministic encoder and the decoder is fed with the latent space to get the reconstruction.

We observe that VAE based models fail at reconstructing SVHN given SVHN. This is especially more visible for product of experts based models (MVAE and MVTCAE. In MLD, the deterministic autoencoders do not suffer from such weakness and achieve overall the best performance.

Figure 13 shows qualitative results for the self-generation. We remark that some samples generated using VAE-based models, the digits differs from the ones in the input sample, indicating information loss due to the latent collapse. For example, in the case of MVAE, generation of the MNIST digit 3, in MVTCAE generation of the SVHN digit 2.

Table 10: Self-generation coherence and quality for **MNIST-SVHN** ( M :MNIST, S: SVHN). The generation quality is measured in terms of FMD for MNIST and FID for SVHN.

| Models | Coherence (%↑) | | | | Quality (↓) | | | |
|---|---|---|---|---|---|---|---|---|
| | $M \to M$ | $M,S \to M$ | $S \to S$ | $M,S \to M$ | $M \to M$ | $M,S \to M$ | $S \to S$ | $M,S \to M$ |
| MVAE | $86.92_{\pm0.8}$ | $88.03_{\pm0.78}$ | $40.62_{\pm0.99}$ | $68.01_{\pm1.29}$ | $10.75_{\pm1.04}$ | $10.79_{\pm1.02}$ | $60.22_{\pm1.01}$ | $59.0_{\pm0.6}$ |
| MMVAE | $87.22_{\pm1.87}$ | $77.35_{\pm4.19}$ | $67.31_{\pm6.93}$ | $39.44_{\pm3.43}$ | $12.15_{\pm1.25}$ | $20.24_{\pm1.04}$ | $58.1_{\pm3.14}$ | $171.42_{\pm4.55}$ |
| MOPOE | $89.95_{\pm0.84}$ | $91.71_{\pm0.77}$ | $67.26_{\pm0.8}$ | $\underline{83.58}_{\pm0.44}$ | $9.39_{\pm0.76}$ | $10.1_{\pm0.73}$ | $53.19_{\pm1.06}$ | $57.34_{\pm1.35}$ |
| NEXUS | $92.63_{\pm0.45}$ | $93.59_{\pm0.4}$ | $\underline{68.31}_{\pm0.46}$ | $83.13_{\pm0.58}$ | $4.92_{\pm0.61}$ | $5.16_{\pm0.59}$ | $85.67_{\pm2.74}$ | $97.86_{\pm2.86}$ |
| MVTCAE | $\underline{94.33}_{\pm0.18}$ | $\underline{95.18}_{\pm0.4}$ | $47.47_{\pm0.76}$ | $\mathbf{86.6}_{\pm0.23}$ | $\underline{4.67}_{\pm0.35}$ | $\underline{4.94}_{\pm0.37}$ | $\underline{52.29}_{\pm1.17}$ | $\underline{53.55}_{\pm1.19}$ |
| MLD | $\mathbf{96.73}_{\pm0.0}$ | $\mathbf{96.73}\pm0.0$ | $\mathbf{82.19}_{\pm0.0}$ | $82.19_{\pm0.0}$ | $\mathbf{2.25}_{\pm0.03}$ | $\mathbf{2.25}\pm0.03$ | $\mathbf{48.47}_{\pm0.63}$ | $\mathbf{48.47}_{\pm0.63}$ |

### E.1.2 DETAILED RESULTS

Table 11: Generative Coherence for **MNIST-SVHN**. We report the detailed version of Table 1 with standard deviation for 5 independent runs with different seeds.

| Models | Coherence (%↑) | | | Quality (↓) | | | |
|---|---|---|---|---|---|---|---|
| | Joint | $M \to S$ | $S \to M$ | Joint(M) | Joint(S) | $M \to S$ | $S \to M$ |
| MVAE | $38.19_{\pm2.27}$ | $48.21_{\pm2.56}$ | $28.57_{\pm1.46}$ | $13.34_{\pm0.93}$ | $68.0_{\pm0.99}$ | $68.9_{\pm1.84}$ | $13.66_{\pm0.95}$ |
| MMVAE | $37.82_{\pm1.19}$ | $11.72_{\pm0.33}$ | $67.55_{\pm9.22}$ | $25.89_{\pm0.46}$ | $146.82_{\pm4.76}$ | $393.33_{\pm4.86}$ | $53.37_{\pm1.87}$ |
| MOPOE | $39.93_{\pm1.54}$ | $12.27_{\pm0.68}$ | $68.82_{\pm0.39}$ | $20.11_{\pm0.96}$ | $129.2_{\pm6.33}$ | $373.73_{\pm26.42}$ | $43.34_{\pm1.72}$ |
| NEXUS | $40.0_{\pm2.74}$ | $16.68_{\pm5.93}$ | $70.67_{\pm0.77}$ | $13.84_{\pm1.41}$ | $98.13_{\pm5.9}$ | $281.28_{\pm16.07}$ | $53.41_{\pm1.54}$ |
| MVTCAE | $48.78_{\pm1}$ | $\underline{81.97}_{\pm0.32}$ | $49.78_{\pm0.88}$ | $12.98_{\pm0.68}$ | $\mathbf{52.92}_{\pm1.39}$ | $69.48_{\pm1.64}$ | $13.55_{\pm0.8}$ |
| MMVAE+ | $17.64_{\pm4.12}$ | $13.23_{\pm4.96}$ | $29.69_{\pm5.08}$ | $26.60_{\pm2.58}$ | $121.77_{\pm37.77}$ | $240.90_{\pm85.74}$ | $35.11_{\pm4.25}$ |
| MMVAE+(K=10) | $41.59_{\pm4.89}$ | $55.3_{\pm9.89}$ | $56.41_{\pm5.37}$ | $19.05_{\pm1.10}$ | $67.13_{\pm4.58}$ | $75.9_{\pm12.91}$ | $18.16_{\pm2.20}$ |
| MLD | $\underline{85.22}_{\pm0.5}$ | $\mathbf{83.79}_{\pm0.62}$ | $\mathbf{79.13}_{\pm0.38}$ | $\underline{3.93}_{\pm0.12}$ | $\underline{56.36}_{\pm1.63}$ | $\mathbf{57.2}_{\pm1.47}$ | $\underline{3.67}_{\pm0.14}$ |

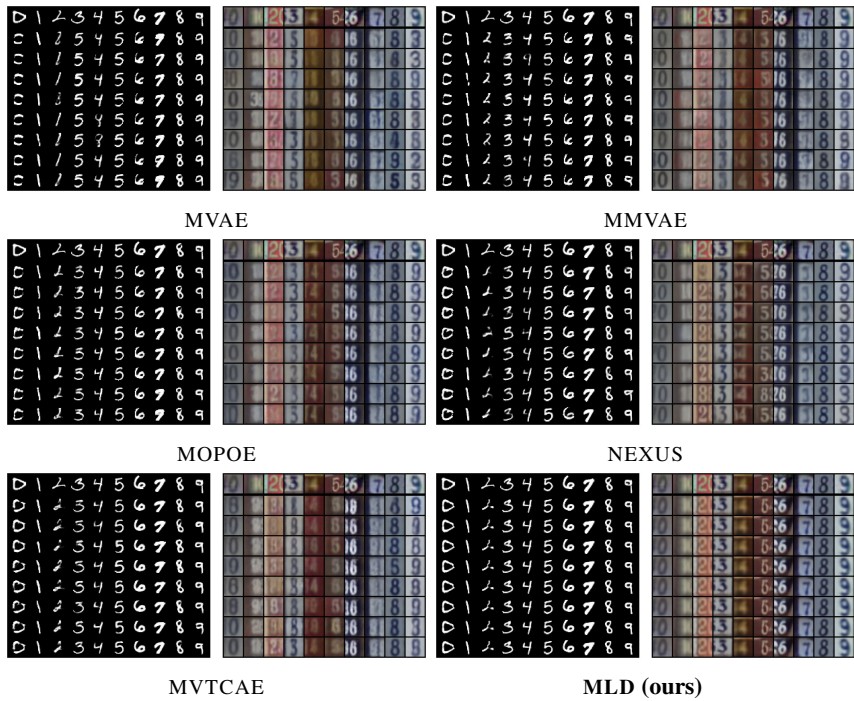

MVAE ... MMVAE

MOPOE ... NEXUS

MVTCAE ... **MLD (ours)**

Figure 13: Self-generation qualitative results for **MNIST-SVHN**. For each model we report: MNIST to MNIST conditional generation in the left, SVHN to SVHN conditional generation in the right.

## E.2 MHD

Table 12: Generative Coherence for **MHD**. We report the detailed version of Table 2 with standard deviation for 5 independent runs with different seeds.

| Models | Joint | I (Image) | | | T (Trajectory) | | | S (Sound) | | |
|---|---|---|---|---|---|---|---|---|---|---|
| | | T | S | T,S | I | S | I,S | I | T | I,T |
| MVAE | $37.77_{\pm 3.32}$ | $11.68_{\pm 0.35}$ | $26.46_{\pm 1.84}$ | $28.4_{\pm 1.47}$ | $95.55_{\pm 1.39}$ | $26.66_{\pm 1.72}$ | $96.58_{\pm 1.06}$ | $58.87_{\pm 4.89}$ | $10.39_{\pm 0.42}$ | $58.16_{\pm 5.24}$ |
| MMVAE | $34.78_{\pm 0.83}$ | $\mathbf{99.7}_{\pm 0.03}$ | $69.69_{\pm 1.66}$ | $84.74_{\pm 0.95}$ | $\underline{99.3}_{\pm 0.07}$ | $85.46_{\pm 1.57}$ | $92.39_{\pm 0.95}$ | $49.95_{\pm 0.79}$ | $50.14_{\pm 0.89}$ | $50.17_{\pm 0.99}$ |
| MOPOE | $48.84_{\pm 0.36}$ | $\underline{99.64}_{\pm 0.08}$ | $68.67_{\pm 2.07}$ | $\underline{99.69}_{\pm 0.04}$ | $99.28_{\pm 0.08}$ | $\underline{87.42}_{\pm 0.41}$ | $99.35_{\pm 0.04}$ | $50.73_{\pm 3.72}$ | $51.5_{\pm 3.52}$ | $56.97_{\pm 6.34}$ |
| NEXUS | $26.56_{\pm 1.71}$ | $94.58_{\pm 0.34}$ | $\underline{83.1}_{\pm 0.74}$ | $95.27_{\pm 0.52}$ | $88.51_{\pm 0.64}$ | $76.82_{\pm 3.63}$ | $93.27_{\pm 0.91}$ | $70.06_{\pm 2.83}$ | $75.84_{\pm 2.53}$ | $89.48_{\pm 3.24}$ |
| MVTCAE | $42.28_{\pm 1.12}$ | $99.54_{\pm 0.07}$ | $72.05_{\pm 0.95}$ | $99.63_{\pm 0.05}$ | $99.22_{\pm 0.08}$ | $72.03_{\pm 0.48}$ | $\underline{99.39}_{\pm 0.02}$ | $\underline{92.58}_{\pm 0.47}$ | $\underline{93.07}_{\pm 0.36}$ | $\underline{94.78}_{\pm 0.25}$ |
| MMVAE+ | $41.67_{\pm 2.3}$ | $98.05_{\pm 0.19}$ | $84.16_{\pm 0.57}$ | $91.88_{\pm}$ | $97.47_{\pm 0.89}$ | $81.16_{\pm 2.24}$ | $89.31_{\pm 1.54}$ | $64.34_{\pm 4.46}$ | $65.42_{\pm 5.42}$ | $64.88_{\pm 4.93}$ |
| MMVAE+(k=10) | $42.60_{\pm 2.5}$ | $99.44_{\pm 0.07}$ | $\mathbf{89.75}_{\pm 0.75}$ | $94.7_{\pm 0.72}$ | $99.44_{\pm 0.18}$ | $\mathbf{89.58}_{\pm 0.4}$ | $95.01_{\pm 0.30}$ | $87.15_{\pm 2.81}$ | $87.99_{\pm 2.55}$ | $87.57_{\pm 2.09}$ |
| MLD | $\mathbf{98.34}_{\pm 0.22}$ | $99.45_{\pm 0.09}$ | $\underline{88.91}_{\pm 0.54}$ | $\mathbf{99.88}_{\pm 0.04}$ | $\mathbf{99.58}_{\pm 0.03}$ | $\underline{88.92}_{\pm 0.53}$ | $\mathbf{99.91}_{\pm 0.02}$ | $\mathbf{97.63}_{\pm 0.14}$ | $\mathbf{97.7}_{\pm 0.34}$ | $\mathbf{98.01}_{\pm 0.21}$ |

Table 13: Generative quality for **MHD**. We report the detailed version of Table 3 with standard deviation for 5 independent runs with different seeds.

| Models | I (Image) | | | | T (Trajectory) | | | | S (Sound) | | | |
|---|---|---|---|---|---|---|---|---|---|---|---|---|
| | Joint | T | S | T,S | Joint | I | S | I,S | Joint | I | T | I,T |
| MVAE | $\underline{94.9}_{\pm 7.37}$ | $93.73_{\pm 5.44}$ | $92.55_{\pm 7.37}$ | $91.08_{\pm 10.24}$ | $39.51_{\pm 6.04}$ | $20.42_{\pm 4.42}$ | $38.77_{\pm 6.29}$ | $19.25_{\pm 4.26}$ | $14.14_{\pm 0.25}$ | $\underline{14.13}_{\pm 0.19}$ | $14.08_{\pm 0.24}$ | $14.17_{\pm 4.26}$ |
| MMVAE | $224.01_{\pm 12.58}$ | $22.6_{\pm 4.3}$ | $789.12_{\pm 12.58}$ | $170.41_{\pm 8.06}$ | $16.52_{\pm 1.17}$ | $\mathbf{0.5}_{\pm 0.05}$ | $30.39_{\pm 1.38}$ | $6.07_{\pm 0.37}$ | $22.8_{\pm 0.39}$ | $22.61_{\pm 0.75}$ | $23.72_{\pm 0.86}$ | $23.01_{\pm 0.67}$ |
| MOPOE | $147.81_{\pm 10.37}$ | $16.29_{\pm 0.85}$ | $838.38_{\pm 10.84}$ | $15.89_{\pm 1.96}$ | $\underline{13.92}_{\pm 0.96}$ | $\underline{0.52}_{\pm 0.12}$ | $33.38_{\pm 1.14}$ | $\mathbf{0.53}_{\pm 0.1}$ | $18.53_{\pm 0.27}$ | $24.11_{\pm 0.4}$ | $24.1_{\pm 0.41}$ | $23.93_{\pm 0.87}$ |
| NEXUS | $281.76_{\pm 12.69}$ | $116.65_{\pm 9.99}$ | $282.34_{\pm 12.69}$ | $117.24_{\pm 8.53}$ | $18.59_{\pm 2.16}$ | $6.67_{\pm 0.23}$ | $33.01_{\pm 3.41}$ | $7.54_{\pm 0.29}$ | $\underline{13.99}_{\pm 0.9}$ | $19.52_{\pm 0.14}$ | $18.71_{\pm 0.24}$ | $16.3_{\pm 0.59}$ |
| MVTCAE | $121.85_{\pm 3.44}$ | $\underline{5.34}_{\pm 0.33}$ | $\underline{54.57}_{\pm 7.79}$ | $\underline{3.16}_{\pm 0.26}$ | $19.49_{\pm 0.67}$ | $0.62_{\pm 0.1}$ | $\underline{13.65}_{\pm 1.24}$ | $0.75_{\pm 0.13}$ | $15.88_{\pm 0.19}$ | $14.22_{\pm 0.27}$ | $\underline{14.02}_{\pm 0.14}$ | $\underline{13.96}_{\pm 0.28}$ |
| MMVAE+ | $97.19_{\pm 12.37}$ | $2.80_{\pm 0.42}$ | $128.56_{\pm 4.47}$ | $114.3_{\pm 11.4}$ | $22.37_{\pm 1.87}$ | $1.21_{\pm 0.22}$ | $21.74_{\pm 3.49}$ | $15.2_{\pm 1.15}$ | $16.12_{\pm 0.40}$ | $17.31_{\pm 0.62}$ | $17.92_{\pm 0.19}$ | $17.56_{\pm 0.48}$ |
| MMVAE+(K=8) | $85.98_{\pm 1.25}$ | $1.83_{\pm 0.26}$ | $70.72_{\pm 1.76}$ | $62.43_{\pm 3.4}$ | $21.10_{\pm 1.25}$ | $1.38_{\pm 0.34}$ | $8.52_{\pm 0.79}$ | $7.22_{\pm 1.6}$ | $14.58_{\pm 0.47}$ | $14.33_{\pm 0.51}$ | $14.34_{\pm 0.42}$ | $14.32_{\pm 0.6}$ |
| **MLD (ours)** | $\mathbf{7.98}_{\pm 1.41}$ | $\mathbf{1.7}_{\pm 0.14}$ | $\mathbf{4.54}_{\pm 0.45}$ | $\mathbf{1.84}_{\pm 0.27}$ | $\mathbf{3.18}_{\pm 0.18}$ | $0.83_{\pm 0.03}$ | $\mathbf{2.07}_{\pm 0.26}$ | $\underline{0.6}_{\pm 0.05}$ | $\mathbf{2.39}_{\pm 0.1}$ | $\mathbf{2.31}_{\pm 0.07}$ | $\mathbf{2.33}_{\pm 0.11}$ | $\mathbf{2.29}_{\pm 0.06}$ |

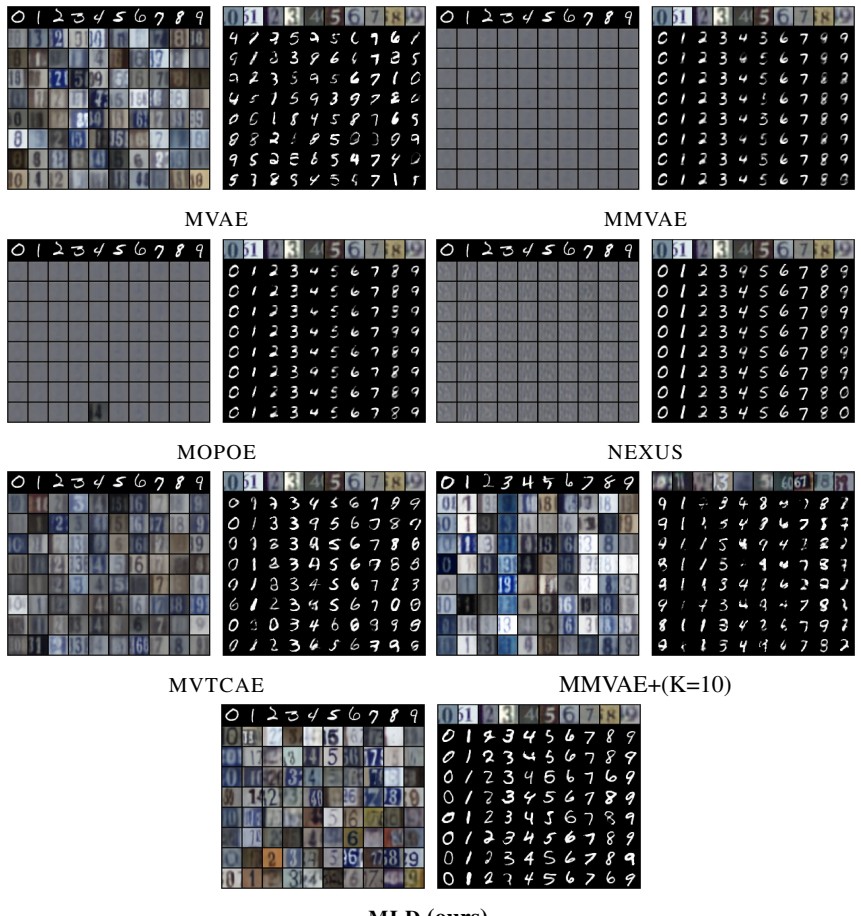

MVAE

MMVAE

MOPOE

NEXUS

MVTCAE

MMVAE+(K=10)

**MLD (ours)**

Figure 14: Additional qualitative results for **MNIST-SVHN**. For each model we report: MNIST to SVHN conditional generation in the left, SVHN to MNIST conditional generation in the right.

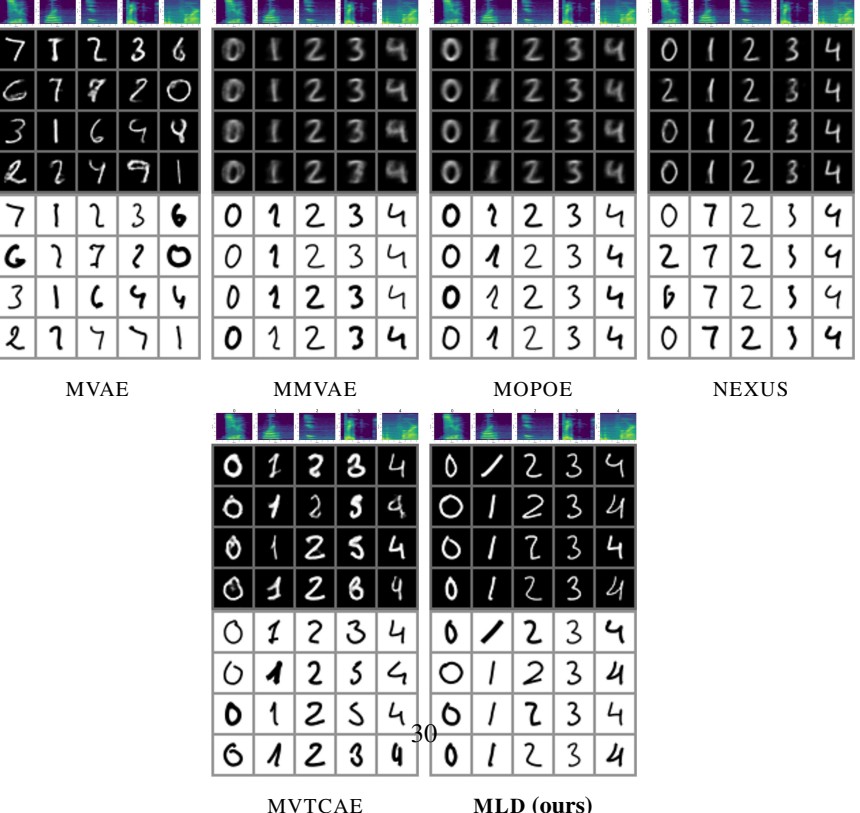

MVAE

MMVAE

MOPOE

NEXUS

MVTCAE

**MLD (ours)**

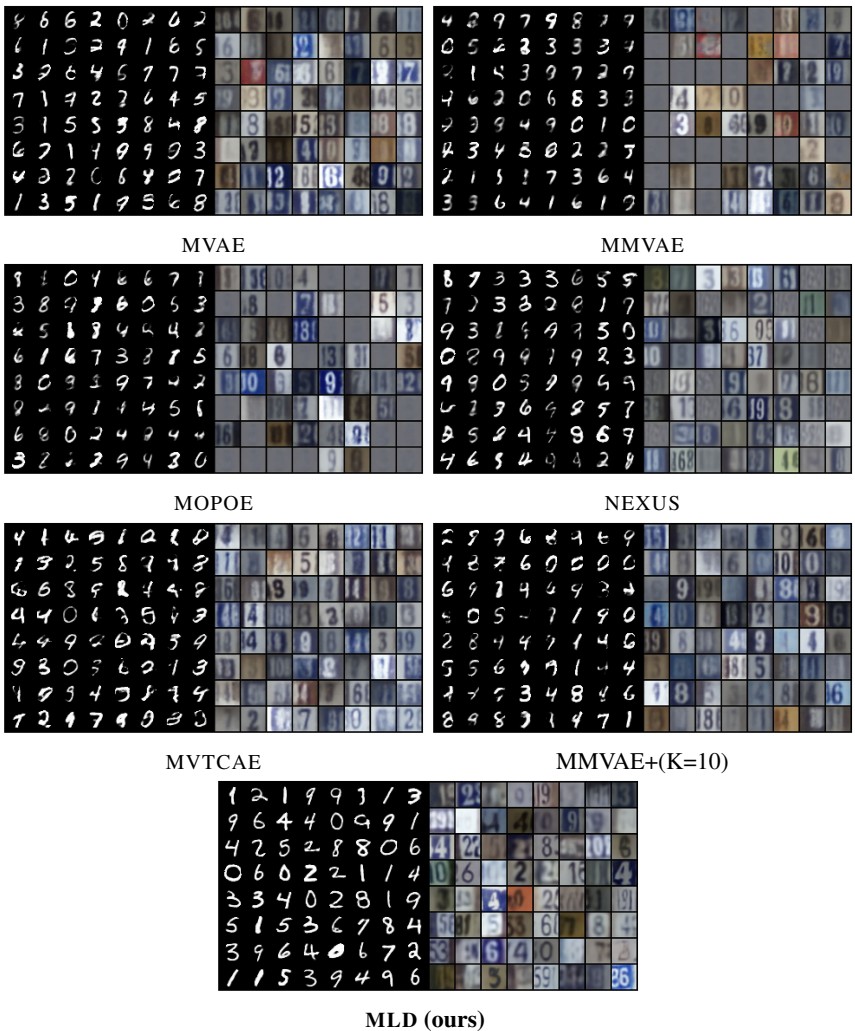

Figure 15: Qualitative results for **MNIST-SVHN** joint generation.

### E.3 POLYMNIST

Table 14: Generation Coherence (%) for **POLYMNIST** (Higher is better) used for the plots in Figure 2 and Figure 9. We report the average *leave one out coherence* as a function of the number of observed modalities. *Joint* refers to random generation of the 5 modalities simultaneously.

| Models | Coherence (%↑) | | | | |
|---|---|---|---|---|---|
| | Joint | 1 | 2 | 3 | 4 |
| MVAE | $4.0_{\pm 1.49}$ | $37.51_{\pm 3.16}$ | $48.06_{\pm 3.55}$ | $53.19_{\pm 3.37}$ | $56.09_{\pm 3.31}$ |
| MMVAE | $25.8_{\pm 1.43}$ | $\mathbf{75.15}_{\pm 2.54}$ | $75.14_{\pm 2.47}$ | $75.09_{\pm 2.6}$ | $75.09_{\pm 2.58}$ |
| MOPOE | $17.32_{\pm 2.47}$ | $\underline{69.37}_{\pm 1.85}$ | $\underline{81.29}_{\pm 2.34}$ | $85.26_{\pm 2.36}$ | $86.7_{\pm 2.39}$ |
| NEXUS | $18.24_{\pm 0.89}$ | $60.61_{\pm 2.51}$ | $72.14_{\pm 2.79}$ | $76.81_{\pm 2.75}$ | $78.92_{\pm 2.64}$ |
| MVTCAE | $0.21_{\pm 0.05}$ | $57.66_{\pm 1.06}$ | $78.44_{\pm 1.31}$ | $\underline{85.97}_{\pm 1.43}$ | $\underline{88.81}_{\pm 1.49}$ |
| MMVAE+ | $26.28_{\pm 2.19}$ | $54.74_{\pm 0.5}$ | $54.06_{\pm 0.33}$ | $55.2_{\pm 1.32}$ | $53.17_{\pm 0.75}$ |
| MMVAE+ (K=10) | $14.53_{\pm 4.94}$ | $58.93_{\pm 6.3}$ | $59.42_{\pm 8.8}$ | $60.77_{\pm 8.03}$ | $58.24_{\pm 7.42}$ |
| MLD IN-PAINT | $\underline{51.65}_{\pm 1.16}$ | $52.85_{\pm 0.23}$ | $77.65_{\pm 0.24}$ | $85.66_{\pm 0.43}$ | $87.29_{\pm 0.29}$ |
| MLD UNI | $48.79_{\pm 0.43}$ | $65.12_{\pm 0.7}$ | $79.52_{\pm 0.8}$ | $82.03_{\pm 1.19}$ | $81.86_{\pm 2.09}$ |
| MLD | $\mathbf{56.23}_{\pm 0.52}$ | $68.58_{\pm 0.72}$ | $\mathbf{84.87}_{\pm 0.19}$ | $\mathbf{88.56}_{\pm 0.12}$ | $\mathbf{89.43}_{\pm 0.27}$ |

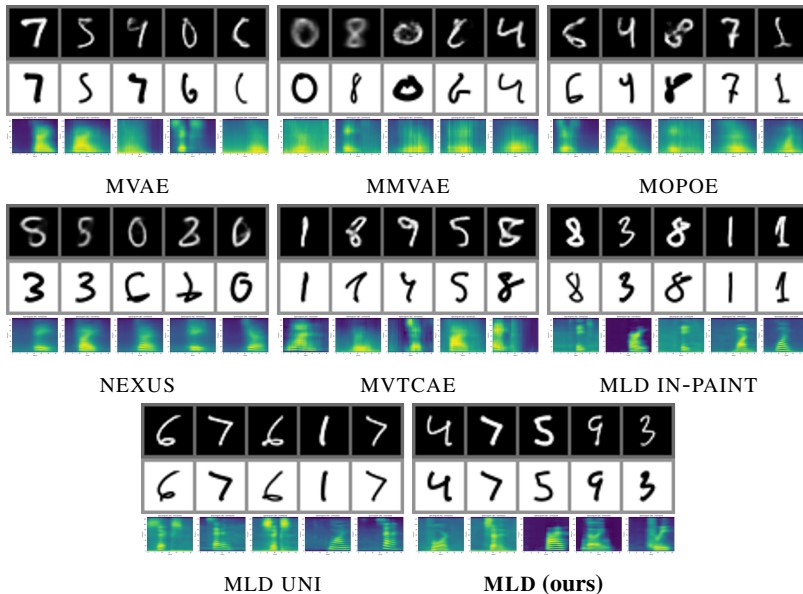

Figure 16: Joint generation qualitative results for **MHD**. The three modalities are randomly generated simultaneously (**Top row**: image,**Middle row**: trajectory vector converted into image, **Bottom row**: sound Mel-Spectogram ).

Table 15: Generation quality (FID ↓) for **POLYMNIST** (lower is better) used for the plots in Figure 2 and Figure 9. Similar to Table 14, we report the average *leave one out* FID as a function of the number of observed modalities. *Joint* refers to random generation quality of the 5 modalities simultaneously.

| Models | Quality (↓) | | | | |
|---|---|---|---|---|---|
| | Joint | 1 | 2 | 3 | 4 |
| MVAE | $108.74_{\pm2.73}$ | $108.06_{\pm2.79}$ | $108.05_{\pm2.73}$ | $108.14_{\pm2.71}$ | $108.18_{\pm2.85}$ |
| MMVAE | $165.74_{\pm5.4}$ | $208.16_{\pm10.41}$ | $207.5_{\pm10.57}$ | $207.35_{\pm10.59}$ | $207.38_{\pm10.58}$ |
| MOPOE | $113.77_{\pm1.62}$ | $173.87_{\pm7.34}$ | $185.06_{\pm10.21}$ | $191.72_{\pm11.26}$ | $196.17_{\pm11.66}$ |
| NEXUS | $91.66_{\pm2.93}$ | $207.14_{\pm7.71}$ | $205.54_{\pm8.6}$ | $204.46_{\pm9.08}$ | $202.43_{\pm9.49}$ |
| MVTCAE | $106.55_{\pm3.83}$ | $78.3_{\pm2.35}$ | $85.55_{\pm2.51}$ | $92.73_{\pm2.65}$ | $99.13_{\pm2.72}$ |
| MMVAE+ | $168.88_{\pm0.12}$ | $165.67_{\pm0.14}$ | $166.5_{\pm0.18}$ | $165.53_{\pm0.55}$ | $165.3_{\pm0.33}$ |
| MMVAE+ (K=10) | $156.55_{\pm3.58}$ | $154.42_{\pm2.73}$ | $153.1_{\pm3.01}$ | $153.06_{\pm2.88}$ | $154.9_{\pm2.9}$ |
| MLD IN-PAINT | $64.78_{\pm0.33}$ | $65.41_{\pm0.43}$ | $65.42_{\pm0.41}$ | $65.52_{\pm0.46}$ | $\underline{65.55}_{\pm0.46}$ |
| MLD UNI | $\mathbf{62.42}_{\pm0.62}$ | $\underline{63.16}_{\pm0.81}$ | $\underline{64.09}_{\pm1.15}$ | $\underline{65.17}_{\pm1.46}$ | $66.46_{\pm2.18}$ |
| MLD | $\underline{63.05}_{\pm0.26}$ | $\mathbf{62.89}_{\pm0.2}$ | $\mathbf{62.53}_{\pm0.21}$ | $\mathbf{62.22}_{\pm0.39}$ | $\mathbf{61.94}_{\pm0.65}$ |

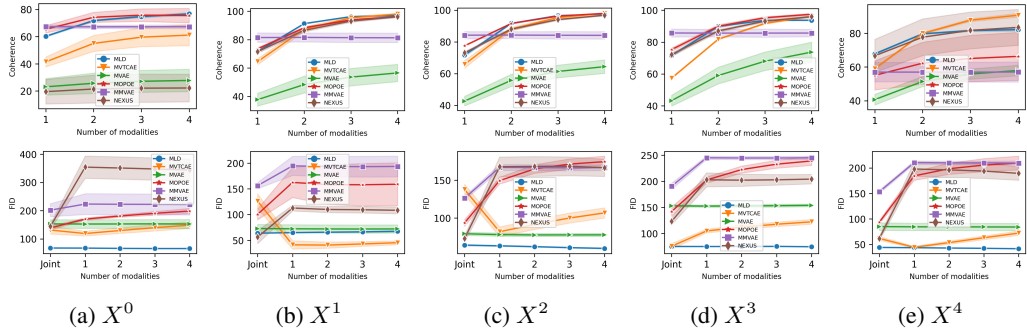

| (a) $X^0$ | (b) $X^1$ | (c) $X^2$ | (d) $X^3$ | (e) $X^4$ |

Figure 18: **Top:** Generation Coherence (%) for **POLYMNIST** (Higher is better). **Bottom:** Generation quality (FID) (Lower is better). We report the average *leave one out* performance as a function of the number of observed modalities for each modality $X^i$. *Joint* refers to random generation of the 5 modalities simultaneously.

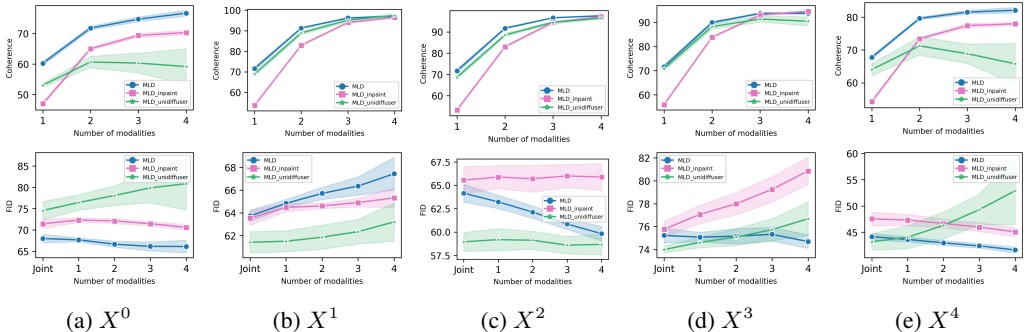

| (a) $X^0$ | (b) $X^1$ | (c) $X^2$ | (d) $X^3$ | (e) $X^4$ |

Figure 19: **Top:** Generation Coherence (%) for **POLYMNIST** (Higher is better).**Bottom:** Generation quality (FID) (Lower is better). We report the average *leave one out* performance as a function of the number of observed modalities for each modality $X^i$. *Joint* refers to random generation of the 5 modalities simultaneously.

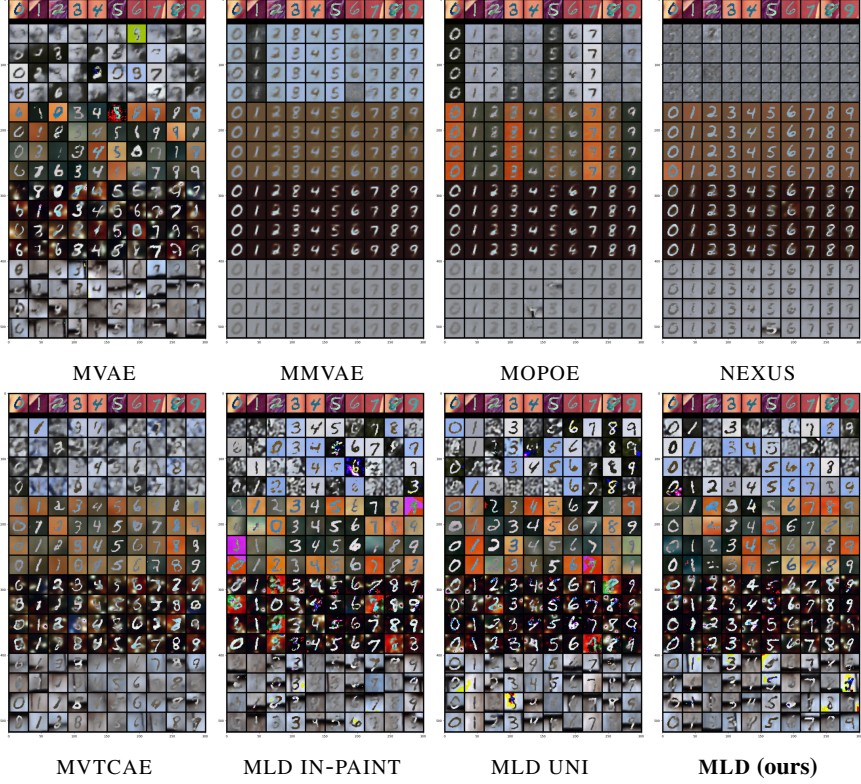

Figure 20: Conditional generation qualitative results for **POLYMNIST** . The modality $X^2$ (dirst row) is used as the condition to generate the 4 remaining modalities(The rows below).

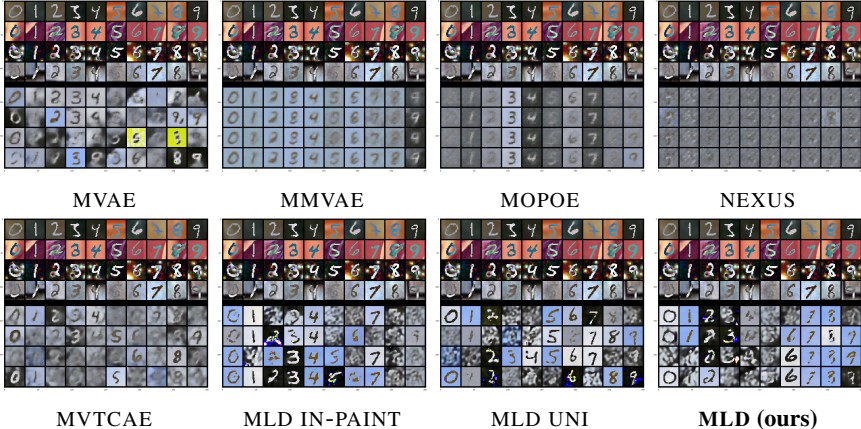

Figure 21: Conditional generation qualitative results for **POLYMNIST**. The subset of modalities $X^1, X^2, X^3, X^4$ (First 4 rows) are used as the condition to generate the modality $X^0$ (The rows below).

### E.3.1 ADDITIONAL EXPERIMENTS WITH PALUMBO ET AL. (2023) ARCHITECTURE

In our experiments on POLYMNIST, we used the same architecture as in Sutter et al. (2021) Hwang et al. (2021) to ensure a fair settings for all the baselines. In Palumbo et al. (2023), the experiments on POLYMNIST are conducted using a different autoencoder architecture which is based on Resnets instead of a a sequence of convolutions layers based autoencoder. We investigate in this section, the performance of MMVAE+ and our MLD using this architecture. For MMVAE+, we keep the same settings as in Palumbo et al. (2023) including the autoencoder architecture, latent size, and importance sampling K=10 with doubly reparameterized gradient estimator (DReG). For MLD, we use the same autoencoder architecture with latent size equal to 160. In Appendix E.3.1, we observe that while the new architecture autoencoder enhance the MMVAE+ performance, our MLD performance is improved as well. Similarly to previous results, MLD achieves simultaneously the best generative coherence and quality.

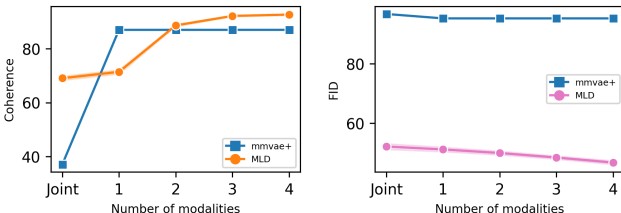

Figure 22: Results for POLYMNIST data-set. Left: a comparison of the generative coherence ( ↑ ) and quality in terms of FID (↓) as a function of the number of inputs.

### E.4 CUB

| Models | Coherence ( ↑ ) | | | Quality ( ↓ ) | |
|---|---|---|---|---|---|
| | Joint | Image → Caption | Caption → Image | Joint → Image | Caption → Image |
| MVAE | 0.66 | **0.70** | 0.64 | 158.91 | 158.88 |
| MMVAE | 0.66 | 0.69 | 0.62 | 277.8 | 212.57 |
| MOPOE | 0.64 | 0.68 | 0.55 | 279.78 | 179.04 |
| NEXUS | 0.65 | 0.69 | 0.59 | 147.96 | 262.9 |
| MVTCAE | 0.65 | **0.70** | 0.65 | 155.75 | 168.17 |
| MMVAE+ | 0.61 | 0.68 | 0.65 | 188.63 | 247.44 |
| MMVAE+(K=10) | 0.63 | 0.68 | 0.62 | 172.21 | 178.88 |
| MLD IN-PAINT | **0.69** | 0.69 | 0.68 | 69.16 | 68.33 |
| MLD UNI | **0.69** | 0.69 | **0.69** | 64.09 | **61.92** |
| MLD | **0.69** | 0.69 | **0.69** | **63.47** | 62.62 |
| MLD* | **0.70** | 0.69 | **0.69** | **22.19** | **22.50** |

Table 16: Generation Coherence (CLIP-S : Higher is better ) and Quality (FID ↓ Lower is better ) for CUB dataset. **MLD\*** denotes the version of our method using a more powerful image autoencoder.

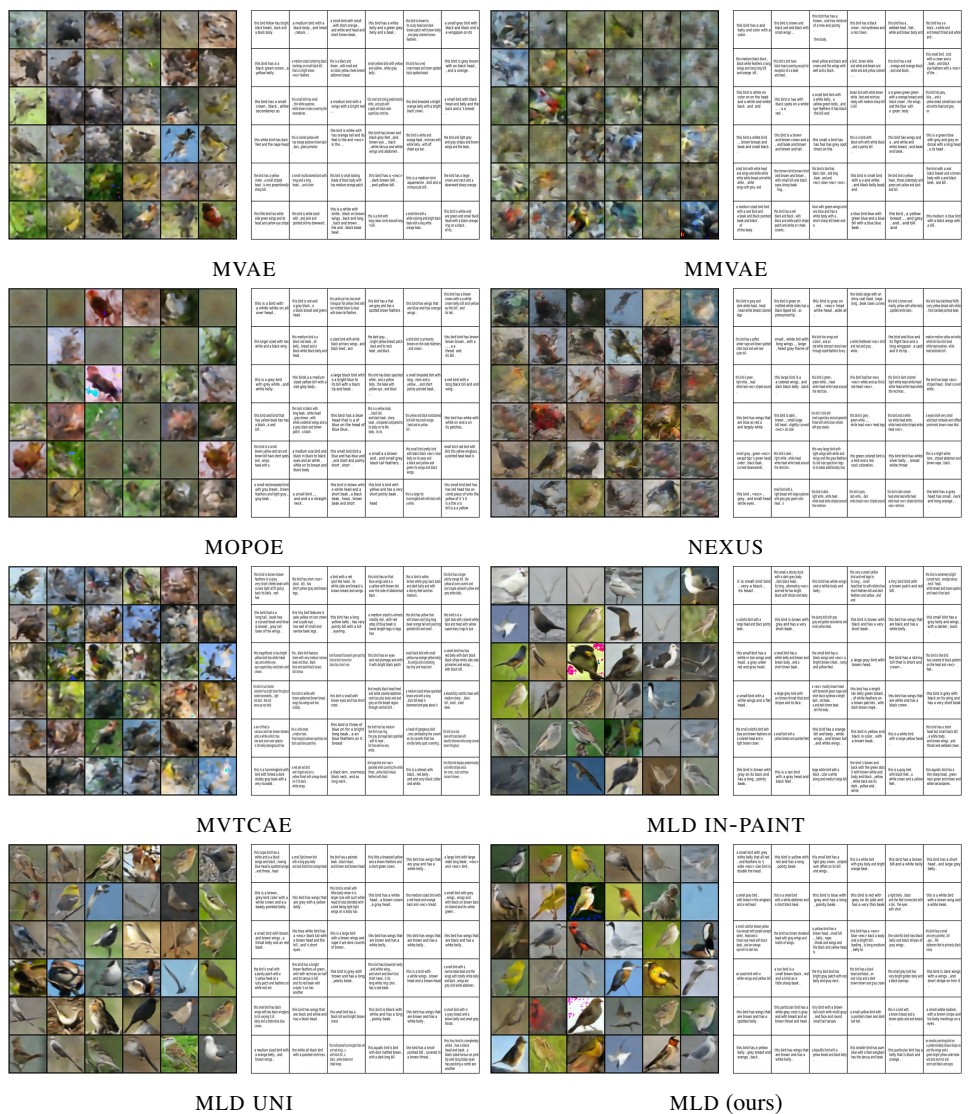

Figure 23: Qualitative results for joint generation on **CUB**.

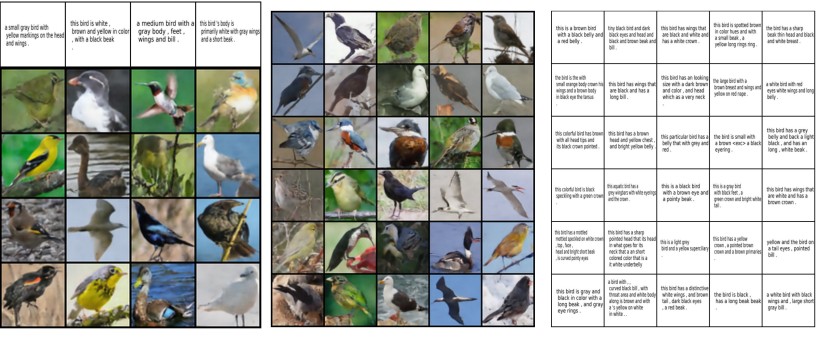

(a) Conditional generation.

(b) Joint generation.

Figure 24: Qualitative results of **MLD\*** on **CUB** data-set with powerful image autoencoder.

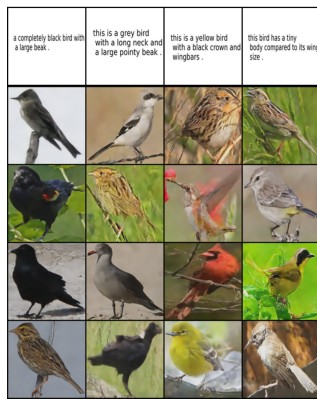

(a) Conditional Generation: Caption used as condition to generate the bird images.

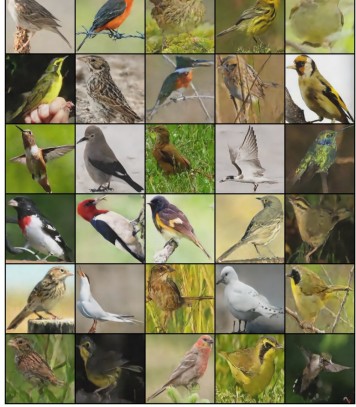

(b) Joint generation: Images and captions are generated simultaneously.

Figure 25: Qualitative results of **MLD\*** on **CUB** data-set with 128x128 resolution with powerful image autoencoder.

## E.5  CELEBAMASK-HQ

In this section, we present additional experiments on the CelebAMask-HQ dataset (Lee et al., 2019a), which consists of face images, each having a segmentation mask, and text attributes, so 3 modalities. We follow the same experimentation protocol as in (Wesego & Rooshenas, 2023) including the autoencoder base architecture. Note that for MLD we use deterministic autoencoders instead of variational autoencoders (Lee et al., 2019a). Similarly, the CelebAMask-HQ dataset is restricted to take into account 18 out of 40 attributes from the original dataset and the images are resized to 128×128 resolution, as done in (Wu & Goodman, 2018; Wesego & Rooshenas, 2023). Please refer to Appendix D, for additional implementation details of MLD.

The Image generation quality is evaluated in terms of FID score. The attributes and the mask having binary values, are evaluated in terms of $F1$ Score against the ground truth. The competitors performance results are reported from Wesego & Rooshenas (2023).

The quantitative results in Table 17 show that MLD outperforms the competitors on the generation quality. It achieves the best F1 score in the attributes generation given Image and Mask modalities. The mask generation best performance is achieved by MOPOE. MLD achieves the second best performance on the mask generation conditioning on both image and attributes modalities. Overall, MLD stands out with the best image quality generation while being on-par with competitors in the mask and attribute generation coherence.

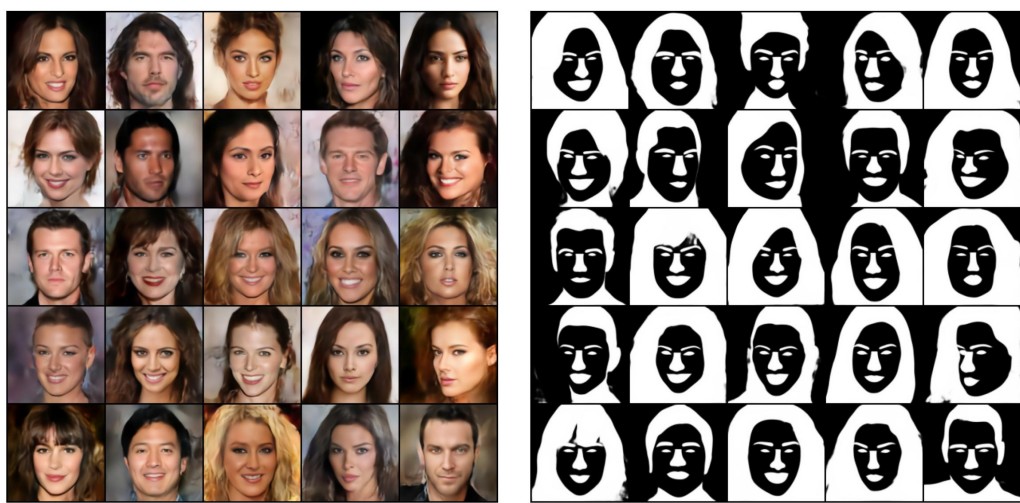

(a) Image

(b) Mask

(c) Attributes

Figure 26: Joint (Unconditional) generation qualitative results of **MLD** on CelebAMask-HQ.

| Models | Attributes | | Image | | | | Mask | |
|---|---|---|---|---|---|---|---|---|
| | Img + Mask F1 | Img F1 | Att + Mask FID | Mask FID | Att FID | Joint FID | Img+Att F1 | Img F1 |
| Wesego & Rooshenas (2023) | | | | | | | | |
| SBM-RAE | 0.62 | 0.6 | 84.9 | 86.4 | 85.6 | 84.2 | 0.83 | 0.82 |
| SBM-RAE-C | 0.66 | 0.64 | 83.6 | 82.8 | 83.1 | 84.2 | 0.83 | 0.82 |
| SBM-VAE | 0.62 | 0.58 | 81.6 | 81.9 | 78.7 | 79.1 | 0.83 | 0.83 |
| SBM-VAE-C | 0.69 | 0.66 | 82.4 | 81.7 | 76.3 | 79.1 | 0.84 | 0.84 |
| MOPOE | 0.68 | **0.71** | 114.9 | 101.1 | 186.8 | 164.8 | 0.85 | **0.92** |
| MVTCAE | 0.71 | 0.69 | 94 | 84.2 | 87.2 | 162.2 | **0.89** | 0.89 |
| MMVAE+ | 0.64 | 0.61 | 133 | 97.3 | 153 | 103.7 | 0.82 | 0.89 |
| Supervised classifier | | 0.79 | | | | | | 0.94 |
| **MLD (ours)** | **0.72** | 0.69 | **52.75** | **51.73** | **53.09** | **54.27** | 0.87 | 0.87 |

Table 17: Quantitative results on CelebAMask-HQ dataset. The performance is measured in terms of FID (↓) and F1 score (↑). The first row is the generated modality while the second row is the modalities used as condition. Supervised classifier designates a classifier performance to predict the attributes or the mask from an image.

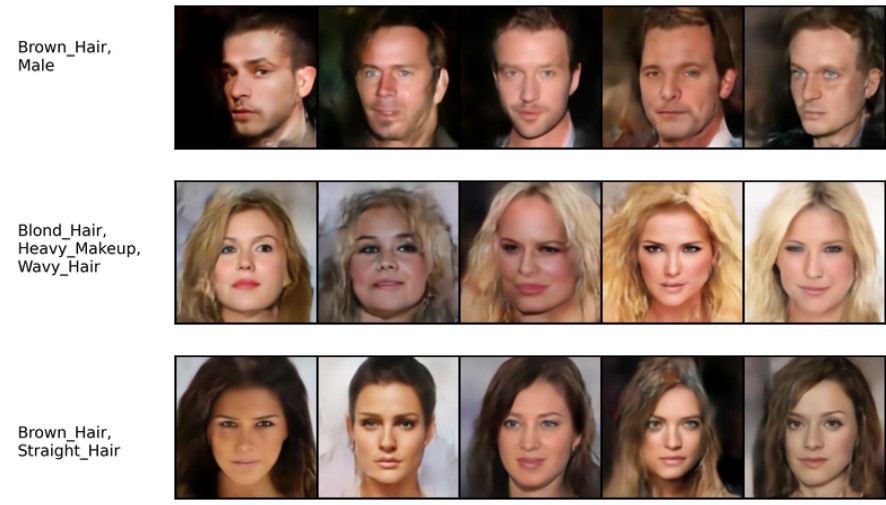

(a) Generated Images

Figure 27: (Attributes → Image ) Conditional generation of **MLD** on CelebAMask-HQ.

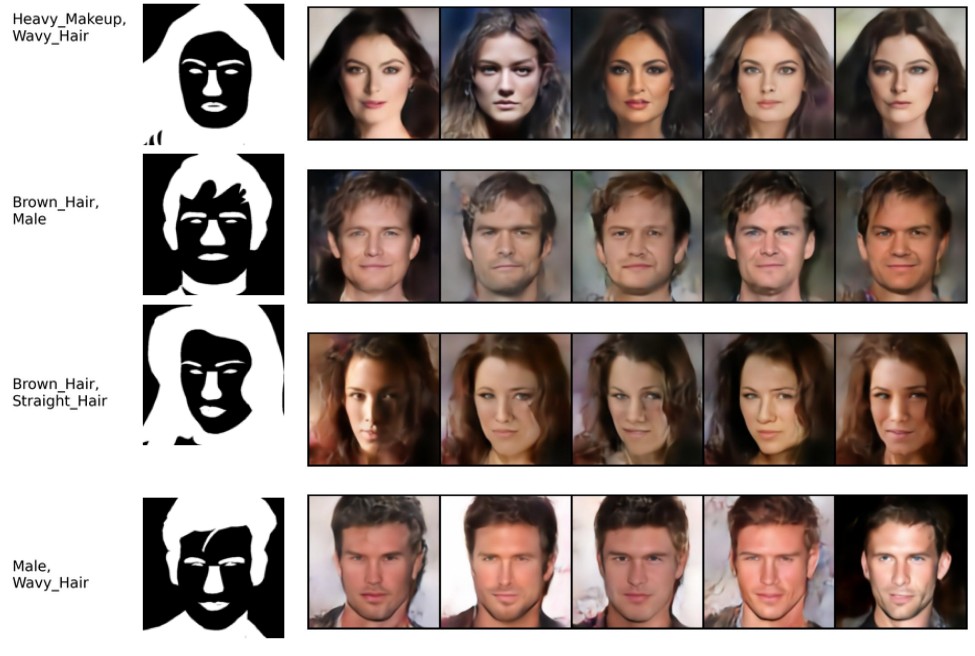

(a) Generated Images

Figure 28: (Attributes,Mask → Image ) Conditional generation of **MLD** on CelebAMask-HQ.

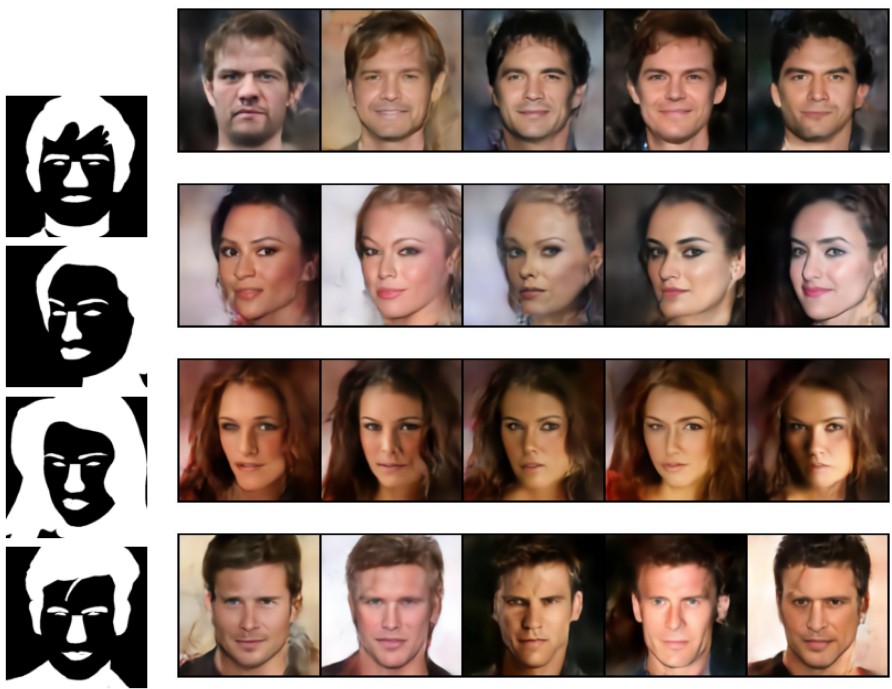

(a) Generated Images

Figure 29: (Mask → Image ) Conditional generation of **MLD** on CelebAMask-HQ.

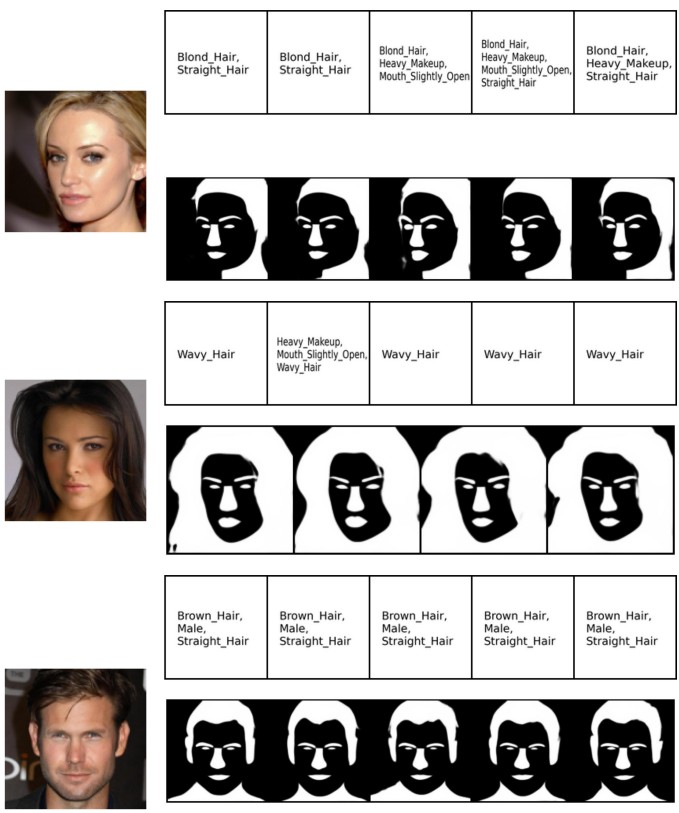

Figure 30: (Image → Attribute,Mask ) Conditional generation of **MLD** on CelebAMask-HQ.

