# OpenReview forum: "Multi-modal Latent Diffusion"
_ICLR.cc/2024/Conference — Submitted to ICLR 2024_

### Official Review · Reviewer_SqcE · 2023-10-27

**Soundness:** 3 good
**Presentation:** 3 good
**Contribution:** 2 fair
**Rating:** 6
**Confidence:** 4

**Summary:**

This paper proposes a new approach to handle the issue, called the coherence-quality tradeoff, of the Multimodal VAE. Specifically, the authors use a set of independently trained, uni-modal, deterministic autoencoders. And, they introduce a multi-time training method to learn the conditional score network in the diffusion model, which enables multi-modal generative modeling.

**Strengths:**

1.	The paper is clear and well-organized.
2.	The appendix significantly enhances the paper by thoroughly supplementing the technical and experimental details.

**Weaknesses:**

1.	It is a nice try to facilitate multi-modal generative modeling by concatenating latent variables from different modalities and employing a diffusion model. However, the methodology of the paper presents numerous issues and lacks in-depth discussion. Furthermore, the method proposed principally depends on intuitive reasoning, with a noticeable absence of solid theoretical underpinnings.

    - The paper employs diffusion on latent variables concatenated from different modalities. One concern is whether the data distributions corresponding to vectors from these various modalities significantly diverge. If so, does utilizing the same noise schedule could result in a lack of synchronization between the denoising and noise-adding processes across the different modalities?
    - Additionally, the optimization speeds of different modalities may inherently vary, leading to discrepancies in performance. In other words, the proposed method might achieve satisfactory results with simplistic datasets, but training becomes substantially more challenging when scaled to extensive, real-world data scenarios.
    - In the context of conditional generation, the authors employ a masking technique to generate the desired modality based on the known one. However, the question arises: how is the intensity of the conditions controlled? This aspect is crucial for ensuring the effectiveness of the generative process.
    - Within the model's framework, a discrepancy arises between the training and inference stages in terms of the number of modalities. For instance, during training, the model might handle three modalities: A, B, and C. However, in a scenario where inference is desired based solely on modality A to predict B, would a masked C still be necessitated? This raises questions about the model's flexibility and its adaptability to accommodate various generative scenarios with different modalities. The capacity to dynamically adjust to these conditions without compromising the integrity of the generative process is pivotal.

2. Some recent work, such as MMVAE+(Palumbo et al., 2023), should be included as a baseline. Work parallel to this paper, score-based multimodal autoencoders (Wesego et al., 2023), should be discussed in Section 2.

3. The experiments require further refinement.

   - The quantitative comparisons on the CUB dataset should be integrated into the main text. Moreover, the coherence metric for image->caption has not improved, necessitating a comprehensive comparative analysis and case demonstrations of caption generation. Additionally, the visual representations in this section of the paper are quite unclear, making them difficult to interpret.
   - Considering that the differences in certain metrics on the CUB dataset are not particularly pronounced, it is recommended to augment the study with a comparative analysis on the Bimodal CelebA dataset.
   - The authors have not made their code available, which makes it hard to reproduce the experimental results.


**Reference:**

[1] Palumbo, Emanuele, Imant Daunhawer, and Julia E. Vogt. "MMVAE+: Enhancing the generative quality of multimodal VAEs without compromises." Fifth Symposium on Advances in Approximate Bayesian Inference-Fast Track. 2023.

[2] Wesego, Daniel, and Amirmohammad Rooshenas. "Score-Based Multimodal Autoencoders." arXiv preprint arXiv:2305.15708 (2023).

**Questions:**

Please refer to the specifics outlined in the “Weaknesses”

---

> ### Author Response · Authors · 2023-11-21
>
> Thank you for your thorough review. We hope to resolve all your concerns with additional clarification and experimental results.
>
> > **It is a nice try to facilitate multi-modal generative modeling by concatenating latent variables from different modalities and employing a diffusion model. However, the methodology of the paper presents numerous issues and lacks in-depth discussion. Furthermore, the method proposed principally depends on intuitive reasoning, with a noticeable absence of solid theoretical underpinnings**
>
> We respectfully disagree with the critique regarding the lack of rigor in the methodology of our paper. Our approach is grounded in more than just intuitive reasoning, please refer to Sections 3, 4 and Appendix A.2. In fact, feedback from other reviewers has pointed out that our work leans heavily towards mathematical rigor, potentially at the expense of intuitive understanding.
>
>
> **The paper employs diffusion on latent variables concatenated from different modalities. One concern is whether the data distributions corresponding to vectors from these various modalities significantly diverge. If so, does utilizing the same noise schedule could result in a lack of synchronization between the denoising and noise-adding processes across the different modalities?**
>
> We can analyze the problem separating two different time regimes: large diffusion times, and small ones.
>
> Large diffusion times **cannot** be the source of synchronization problems. Indeed it is widely known that under loose assumptions [Collet2008], the discrepancy (in KL sense) between time varying probability densities and their final (Gaussian in this case) steady state distributions decreases exponentially fast. In the language of our work: $KL(q(r^{A_1},t) | \rho(r^{A_1})\leq C\exp(-\lambda t)$, where the parameters $C,\lambda>0$ are independent of time. This result holds for any starting distribution $q(r^{A_1},0)$ [Collet2008].
>
> For small diffusion times, the question of whether different modalities respond differently to noise injection is important. In Appendix A.2, “Latent space robustness against diffusion perturbation” of the submitted paper, we discuss in length about this matter (see also Figure 5).
>
> It is important to notice that it is this very question, together with the information theoretic analysis presented in Section 4.2, that motivates the multi-time diffusion process as an alternative to the inpainting approach (which is based on a single diffusion time).
>
> * [Collet2008] J.F. Collet and F. Malrieu. Logarithmic sobolev inequalities for inhomogeneous markov semigroups. ESAIM: Probability and Statistics, 12:492–504, 2008
>
> >**Additionally, the optimization speeds of different modalities may inherently vary, leading to discrepancies in performance. In other words, the proposed method might achieve satisfactory results with simplistic datasets, but training becomes substantially more challenging when scaled to extensive, real-world data scenarios.**
>
> We respectfully disagree, this concern is speculative. In our experiments, we have not observed any issues related to the optimization speed raised by different modalities. The performance of our method has remained robust and consistent across various datasets.
>
> As explained in the submitted version of the paper, we “ avoid any form of interference in the back-propagated gradients corresponding to the uni-modal reconstruction losses. Consequently, gradient conflict issues [Javaloy et al., 2022], where stronger modalities pollute weaker ones, are avoided.” (see Section 3, below Equation 2 in our paper).
>
> The only impact of scaling to larger, higher-resolution datasets is on the computational cost of the (single, amortized) score network, which scales with the dimensions of the concatenated latent sizes. However, this does not inherently make training more challenging from a methodological standpoint.
>
> **[Note: this reply continues in a separate comment.]**

---

> ### Author Response · Authors · 2023-11-21
> **.**
>
> > **In the context of conditional generation, the authors employ a masking technique to generate the desired modality based on the known one. However, the question arises: how is the intensity of the conditions controlled? This aspect is crucial for ensuring the effectiveness of the generative process.**
>
> In our MLD method, conditional generation is indeed obtained through a masking process. This allows, assuming for example to consider as conditioning modality $A=a$, to have access to the *conditional score* of the diffused modality $B$ for any given diffusion time $t\in[0,T]$. Consequently, it is possible to generate samples from $p(B|A=a)$, by simulating the reverse diffusion.
> In the our implementation, there is not such a thing such as the *intensity*  of the conditions, as such strength assumes only binary values: either we generate conditionally, or unconditionally. This however, can be easily relaxed using the same simple tricks described in [Ho2021], where the introduction of an extra guidance parameter allows modulation of the importance of the conditioning modality. While interesting, we consider such a simple case an interesting venue for future works.
>
>
> [Ho2021] J. Ho, T. Saliman. Classifier-free Diffusion Guidance, NeurIPS 2021 Workshop on Deep Generative Models and Downstream Applications.
>
> >**Within the model's framework, a discrepancy arises between the training and inference stages in terms of the number of modalities. For instance, during training, the model might handle three modalities: A, B, and C. However, in a scenario where inference is desired based solely on modality A to predict B, would a masked C still be necessitated? This raises questions about the model's flexibility and its adaptability to accommodate various generative scenarios with different modalities. The capacity to dynamically adjust to these conditions without compromising the integrity of the generative process is pivotal.**
>
>
> In the scenario you describe, where the model is trained on three modalities (A, B, and C) but inference is desired based on modality A to “predict” B, it is not necessary to involve modality C. If the goal is to ensure that modality C does not influence the generation from A to B, our method simply considers C as a missing modality during the inference stage. This flexibility is an integral part of our model's design, ensuring that it can dynamically adjust to different conditions without compromising the integrity of the generative process.
>
> >**Some recent work, such as MMVAE+(Palumbo et al., 2023), should be included as a baseline. Work parallel to this paper, score-based multimodal autoencoders (Wesego et al., 2023), should be discussed in Section 2.**
>
>
> We thank the reviewer for the suggestions. We included in the new version of the full experimental campaign results for all the datasets using MMVAE+ [Palumbo et al., 2023]. Our findings suggest that MLD outperforms this additional competitor, corroborating the claim that our method has excellent performance compared to the current SoA.
>
> We also now cite the work from Wesego et al., 2023, but prefer to refer to it upfront in Section 1, rather than discussing it as part of the limitations of multimodal VAEs. Note that the work from Wesego et al. 2023, is an interesting approach that only loosely resembles our method. To the best of our understanding: 1) it uses Variational autoencoders, to produce latent variables of the same size for all modalities; 2) conditional generation is achieved through an auxiliary energy based model in the spirit of a “classifier-free guidance” mechanism. We have performed additional experiments using the very same dataset in Wesego et al., 2023 (despite it being a concurrent submission) and included results in our new version of the paper: we did not have time to re-implement their method from scratch, so we used the results reported in their tables and compared them to our results. Our findings indicate that our method overall achieves the best results in terms of generative quality and coherence metric.
>
> [Wesego2023] Daniel Wesego and Amirmohammad Rooshenas, Score-Based Multimodal Autoencoders. https://arxiv.org/abs/2305.15708
>
> **[Note: this reply continues in a separate comment.]**

---

> > ### Author Response · Authors · 2023-11-21
> > **.**
> >
> > >**The quantitative comparisons on the CUB dataset should be integrated into the main text. Moreover, the coherence metric for image->caption has not improved, necessitating a comprehensive comparative analysis and case demonstrations of caption generation. Additionally, the visual representations in this section of the paper are quite unclear, making them difficult to interpret.**
> >
> > Due to space limitations, we cannot move the CUB experiments to the main. The reviewer indicates that results in table 16, Appendix E.4 show that in one particular case, image->caption, the coherence of MLD is not the best. A careful look at the table shows that the performance of MLD in this task is on par with alternatives, and better for all other tasks (caption-> image, joint quality, caption->image quality).
> >
> > Image to caption is a very difficult task, for which even specialized models [Li2023] relying on LLMs require metrics that only loosely measure coherency, and prompt hints that instill information for text generation. We believe this is an exciting area of research, but it is outside the scope of our objectives in this paper.
> >
> > The visual representations we display in the main and appendix for CUB images are in-line with the results obtained by all the works we compare to ( see fig. 7 of [Palumbo2023], see fig 4 of [Daunhawer2022]). Simply put, image generation quality suffers heavily in most of the competitors, and when it does not, coherency is very bad. This is exactly the quality/coherence tradeoff that has been pointed out in the literature and that we study in our work.
> >
> > * [Li2023] J. Li, et al. BLIP-2: Bootstrapping Language-Image Pre-training with Frozen Image Encoders and Large Language Models. https://arxiv.org/abs/2301.12597
> > * [Palumbo2023] E. Palumbo, et al. MMVAE+: Enhancing the Generative Quality of Multimodal VAEs Without Compromises, ICLR 2023, https://openreview.net/pdf?id=BYHy9WwxFU
> > * [Daunhawer2022] Imant Daunhawer, et al. On the Limitations of Multimodal VAEs, ICLR 2022, https://arxiv.org/pdf/2110.04121.pdf
> >
> > > **Considering that the differences in certain metrics on the CUB dataset are not particularly pronounced, it is recommended to augment the study with a comparative analysis on the Bimodal CelebA dataset.**
> >
> > Thank you for the suggestion! We performed additional experiments (see Appendix E.5) with the CelebAMask-HQ dataset [Lee2020] as done in [Wesego2023], which consists of face images, each having a segmentation mask, and attributes, so 3 modalities. Our results are clear: the method we propose outperforms competitors from the state of the art, even with this new dataset.
> >
> > * [Lee2020] Cheng-Han Lee, Ziwei Liu, Lingyun Wu, and Ping Luo. Maskgan: Towards diverse and interactive facial image manipulation. In IEEE Conference on Computer Vision and Pattern Recognition (CVPR), 2020.
> > * [Wesego2023] Daniel Wesego and Amirmohammad Rooshenas, Score-Based Multimodal Autoencoders. https://arxiv.org/abs/2305.15708
> >
> > >**The authors have not made their code available, which makes it hard to reproduce the experimental results.**
> >
> > We have now made code available by uploading it as an anonymous zip supplement.

---

> > > ### Comment · Reviewer_SqcE · 2023-11-22
> > >
> > > Thank you for your feedback. Upon thorough review of the revised submission, I have chosen to increase my rating.

---

> > > > ### Author Response · Authors · 2023-11-23
> > > > **Thank you**
> > > >
> > > > Dear Reviewer, thank you for taking time to read our rebuttal, and for revising your initial assessment of our work.

---

### Official Review · Reviewer_865d · 2023-10-30

**Soundness:** 3 good
**Presentation:** 2 fair
**Contribution:** 3 good
**Rating:** 5
**Confidence:** 3

**Summary:**

The authors present a method for conditionally generating multiple modalities of data which allows for certain modalities to be generated while conditioned on other existing modalities (e.g. generating images and audio from text). In contrast with VAE-based approaches, which can suffer from information loss due to the explicit separation of distinct modality subsets, the authors propose their method MLD, which avoids the problem by training separate unimodal autoencoders _deterministically_, and allowing a diffusion model to learn conditional generation of each modality’s latent space. The diffusion model is trained to be conditioned on random subsets of modalities so that it remains robust to conditioning on any subset. The authors then compare MLD on several datasets against multi-modal VAE approaches.

**Strengths:**

### Good comparisons to multi-model VAEs with encouraging results

The authors do a good job of comparing their method MLD to other VAE-based works, and their results on their datasets are encouraging.

### Good comparison in-painting and explanation for why it works less well

The authors also preemptively address a very natural question of why in-painting and cold diffusion might work less well compared to their approach of robustly training the diffusion model to be conditioned on different subsets of modalities. Their ablation study empirically justifies this claim, or at least it shows that in-painting is not significantly better than their proposed method.

**Weaknesses:**

### No comparison to multi-modal diffusion models

Although the authors have done a good job comparing to previous VAE-related works for multi-modal generation, there is no comparison with purely diffusion-based works. For example, the rather popular Any-to-Any Composable Diffusion (CoDi) (Tang, et. al., 2023) work is very closely related to MLD. CoDi also attempts to solve the multi-modal generation problem. Like the proposed method, CoDi performs diffusion in latent space and allows for conditioning on arbitrary subsets of modalities. Conditioning is done through “latent alignments”, where the latent space of some modalities are attended to by generated modalities. To tackle the problem of coherence, CoDi performs “bridge alignments” to pre-align the latent representations of each modality. Since this work is so similar in methodology to the proposed work here, it should be benchmarked against, as well.

### Datasets benchmarked against are somewhat limited

The datasets in this work are fairly small (the MNIST datasets). The CUB dataset is larger, but only has two modalities. Since one of the core claims of this paper is successful multi-modal generation for arbitrary _subsets_ of modalities, this paper would be much stronger if it could also show MLD working well on another large dataset with more than two modalities (e.g. videos with audio and text).

### More background on multi-modal VAEs would be nice

Diffusion models are fairly common and well known at this point, but multi-modal VAEs are less well known (in my opinion). It would have been nice to have more background on how multi-modal VAEs work before describing their limitations.

A main figure which illustrates the structure of these multi-modal VAEs in comparison to the proposed method would also be very helpful.

### Some of the equations and math could be clearer

Oftentimes, there are equations presented which are presented without much explanation of each component (e.g. Equation 4, Equation 6, Equation 7). These can be a bit confusing to go through when there are simple English descriptions that can be offered instead (or certainly in conjunction) (e.g. “keeping the modalities in $A_1$ static throughout the forward and reverse diffusion process”). These equations should be explained in more straightforward English or even replaced with English descriptions, because the equations do not aid in additional understanding of the paper’s contributions.  Other equations like Equation 5 are certainly not needed for the understanding of this paper, since the modification onto diffusion-model training is very minor.

**Questions:**

Minor grammatical suggestion: there should be an en-dash in “coherence–quality tradeoff”, not a hyphen.

---

> ### Author Response · Authors · 2023-11-21
>
> > **Good comparisons to multi-model VAEs with encouraging results, Good comparison in-painting and explanation for why it works less well**
>
> We thank the reviewer  for acknowledging the two main strengths of our work: a comprehensive experimental validation, which shows superior performance of MLD compared to competitors, and the need to modify classical diffusion processes to include the concept of multi-time, being the naive approach based on in-painting limited, as argued using information theoretic principles.
>
> > **Although the authors have done a good job comparing to previous VAE-related works for multi-modal generation, there is no comparison with purely diffusion-based works.**
>
> Thank you for the suggestion, we know the work of Tang et al. very well! Let us start by re-stating the objective of our work: study the limitations of a large literature on multimodal generative modeling through latent fusion, and propose a radically new approach to latent mixing to overcome the limitations exemplified by the generation quality/coherence traderoff. To do that, we need to be able to compute quality (easy) and coherence (not so easy).
> Now, we indeed considered using CoDi in our experimental campaign. We see two options:
>
> 1) Use the datasets from CoDi. For MLD, this requires training four autoencoders and one diffusion model on data that is several orders of magnitude larger than what our computational infrastructure allows (in CoDi, they use pre-trained models, we can’t do the same, because our method is conceptually different). Even if we were up to the challenge, we would face a greater problem: computing the coherence between video modality and another set of modalities is not well understood (e.g., see Table 10 of CoDi, where coherence between a video input and any subset of output is not available).
>
> 2) Use the datasets we adopt in our work, which would allow comparing CoDi not only to our MLD, but also to all the other methods we study. In this case, pre-trained CoDi diffusion models cannot be used, and we would have to re-train all components of CoDI from scratch. Even if we were up to the challenge, we would face a greater problem: CoDi **requires** a “master modality”, which is text. Many of the datasets we use (see also our answer to Reviewer yi7k on real-world applications) do not have the text modality, and it is not clear how to select a “master modality” to align to.
>
> To conclude, while we recognize CoDi's impressive engineering and its contributions to the field (and we properly cite this work), its primary focus on text-driven-multimodal applications is not compatible with our goal of studying "latent fusion" and the quality/coherence tradeoff.
>
> >**Datasets benchmarked against are somewhat limited his paper would be much stronger if it could also show MLD working well on another large dataset with more than two modalities (e.g. videos with audio and text).**
>
> We understand that the video modality is attractive, especially from the aesthetic point of view. Pushing the limits by striving for higher and higher resolutions is key for application-oriented methods that aim at surpassing SOTA results.
>
> However, our work focuses on a different research question: is it possible to break the limits of multimodal latent fusion with a radically new approach to achieve a sweet spot in the generative quality/coherence tradeoff? The answer is affirmative, and our claims are fully supported by the results obtained with our experimental protocol.
>
> In light of a similar suggestion from Reviewer SqcE, we performed additional experiments (see Appendix E.5) with the CelebAMask-HQ dataset [Lee2020] as done in [Wesego2023], which consists of face images, each having a segmentation mask, and attributes, so 3 modalities. Our results are clear: the method we propose outperforms competitors from the state of the art, even with this new dataset.
> Other kinds of datasets, for example those including videos, are not appropriate for our goal because computing the coherence metric is not currently possible.
>
> * [Lee2020] Cheng-Han Lee, Ziwei Liu, Lingyun Wu, and Ping Luo. Maskgan: Towards diverse and interactive facial image manipulation. In IEEE Conference on Computer Vision and Pattern Recognition (CVPR), 2020.
> * [Wesego2023] Daniel Wesego and Amirmohammad Rooshenas, Score-Based Multimodal Autoencoders. https://arxiv.org/abs/2305.15708
>
>
> **[Note: this reply continues in a separate comment.]**

---

> > ### Author Response · Authors · 2023-11-21
> > **.**
> >
> > >**More background on multi-modal VAEs would be nice. A main figure which illustrates the structure of these multi-modal VAEs in comparison to the proposed method would also be very helpful.**
> >
> > Thank you for the suggestion! We can commit to prepare a short overview of the literature on multimodal VAEs, and a schematic representation thereof, as an additional section in the Appendix, for the camera ready version of this paper. We will cover: MVAE, which uses a product of expert latent fusion, MMVAE, which uses a mixture of expert latent fusion, and the recent MMVAE+ which is the state of the art (as of ICLR 2023) method in this area.
> >
> > >**Some of the equations and math could be clearer**
> >
> > To align with the reviewer’s request, we can commit to writing a new section in the Appendix called “Intuitive summary” which we hope will help the readers grasp at an intuitive level the key ideas of our method, without using the language of mathematics. Please refer to Appendix A.5 in the new version of the paper for a preliminary draft of such a discussion.
> >
> >
> > >**Minor grammatical suggestion: there should be an en-dash in “coherence–quality tradeoff”, not a hyphen.**
> >
> > Thank you!, we fixed it.

---

> > > ### Comment · Reviewer_865d · 2023-11-22
> > >
> > > Thank you to the authors for providing additional details.
> > >
> > > **Comparison with CoDi**
> > >
> > > The explanation for why it would be prohibitive to apply MLD to the exact datasets from CoDi makes sense. It is unfortunate that the method does not scale to such larger datasets, and that is likely a drawback of MLD which should be made clear (which is okay).
> > >
> > > However, comparing with CoDi as a method would still be very informative to understand how MLD works compared to another diffusion-based method. The two methods are fundamentally similar, and especially given how well-know CoDi is, it is important to show some comparison with CoDi (even if MLD underperforms in some areas compared to CoDi). The selection of master modality can be done fairly easily, too. For example, for MHD, using the image modality as the master would be a reasonable selection just for benchmarking purposes.
> > >
> > > **Additional datasets**
> > >
> > > The new results on the additional datasets are quite nice and have helped alleviate this particular concern for me. It would still be very informative, I think, to see how MLD performs on datasets with more modalities, as MLD is somewhat unique due to how it effectively trains (and samples from) the diffusion model on arbitrary partitions of the modalities. That said, it would understandable if there simply aren't any real-world applications which have many modalities.
> > >
> > > However, something that has come up in this discussion (and was touched upon in my initial review) is the question of scalability. Multi-modal models will generally always be more difficult and expensive to train, but the method would still need to be able to scale relatively well to be applied to real-world situations (even if it is infeasible to apply it to the CoDi datasets in such a short time frame, which is understandable). Can the authors provide some understanding of the training time (in number of epochs and wall time), comparing the VAE and diffusion-model training time for MLD (I see the number of epochs in Tables 8 and 9) versus the training time of a simple multi-modal diffusion model (which very well may underperform in both quality and coherence, and can only generate all modalities of an object unconditionally)?

---

> > > > ### Author Response · Authors · 2023-11-22
> > > > **Thank you for your comments and for reading our rebuttal!**
> > > >
> > > > Dear Reviewer 865d,
> > > > we would like to thank you for your work and time in reading our rebuttal, and for engaging with us!
> > > > Here are some additional remarks to your questions.
> > > >
> > > > >The explanation for why it would be prohibitive to apply MLD to the exact datasets from CoDi makes sense. It is unfortunate that the method does not scale to such larger datasets, and that is likely a drawback of MLD which should be made clear (which is okay).
> > > >
> > > > We never said that MLD cannot scale to such larger datasets! The problem is that our computational infrastructure does not allow us to perform such experiments in one week! This is not a drawback of MLD, it is just a limitation of the infrastructure available in our lab.
> > > >
> > > > > However, comparing with CoDi as a method would still be very informative to understand how MLD works compared to another diffusion-based method. The two methods are fundamentally similar, and especially given how well-know CoDi is, it is important to show some comparison with CoDi (even if MLD underperforms in some areas compared to CoDi). The selection of master modality can be done fairly easily, too. For example, for MHD, using the image modality as the master would be a reasonable selection just for benchmarking purposes.
> > > >
> > > > This is a fair proposal. We can commit to comparing MLD and CoDi on a dataset such as MHD for a possible camera ready version of the paper. We would just like to point out that MLD and CoDi have a key difference: MLD does not need cross-attention and does not rely on conditioning signals as CoDi, but instead relies on a joint diffusion model, that by design allows (latent) modalities to influence each other.
> > > >
> > > > > The new results on the additional datasets are quite nice and have helped alleviate this particular concern for me. It would still be very informative, I think, to see how MLD performs on datasets with more modalities, as MLD is somewhat unique due to how it effectively trains (and samples from) the diffusion model on arbitrary partitions of the modalities. That said, it would understandable if there simply aren't any real-world applications which have many modalities.
> > > >
> > > > Yes, we agree with the reviewer: for many real-world applications (even the ones we have considered as discussed in our rebuttal to Reviewer yi7k) the number of modalities seem to be limited. We are not sure to understand if the Reviewer is inquiring about the scalability of MLD to a very large number of modalities, or if the Reviewer is more interested in “variety” of modalities.
> > > >
> > > > > However, something that has come up in this discussion (and was touched upon in my initial review) is the question of scalability. Multi-modal models will generally always be more difficult and expensive to train, but the method would still need to be able to scale relatively well to be applied to real-world situations (even if it is infeasible to apply it to the CoDi datasets in such a short time frame, which is understandable). Can the authors provide some understanding of the training time (in number of epochs and wall time), comparing the VAE and diffusion-model training time for MLD (I see the number of epochs in Tables 8 and 9) versus the training time of a simple multi-modal diffusion model (which very well may underperform in both quality and coherence, and can only generate all modalities of an object unconditionally)?
> > > >
> > > > Thank you for the question. MLD requires:
> > > >
> > > > * training of a AE (not a VAE) per modality: the cost is a function of the architecture complexity. In our work, we have used the same architecture when comparing all methods.
> > > > * training of a diffusion model: in our work we use a score network implemented with a simple MLP with residual blocks, which makes our approach faster to train compared to diffusion models that use large and complex U-Nets.
> > > >
> > > > As a demonstration of the efficiency of our MLD, we could run a whole new set of experiments on the requested CelebAMask-HQ dataset over the weekend! As pointed out by the reviewer, we already report the number of epochs used for training in Tab 8 and 9, which is a metric that is independent of the hardware available for training.
> > > >
> > > > Thank you very much for your remarks, we hope that we have further clarified your doubts!

---

### Official Review · Reviewer_yi7k · 2023-10-30

**Soundness:** 3 good
**Presentation:** 2 fair
**Contribution:** 3 good
**Rating:** 6
**Confidence:** 4

**Summary:**

The paper addresses the challenges associated with multi-modal generative modeling, a domain that focuses on generating data across multiple modalities such as images, text, and audio. The primary concern is the coherence-quality tradeoff observed in existing Multi-modal Variational Autoencoders (VAEs), where improving generative coherence across modalities might compromise the generation quality and vice versa.
To tackle these challenges, the authors introduce a novel approach named Multimodal Latent Diffusion (MLD). Unlike traditional multi-modal VAEs that often suffer from latent collapse and information loss, MLD employs independently trained, deterministic uni-modal autoencoders. Each modality is encoded into a specific latent variable, and these variables are then concatenated. The joint data generation is facilitated by a score-based diffusion model in the latent space, which reverses a stochastic noising process starting from a Gaussian distribution.
The experimental results are promising, outperforming the baselines.

**Strengths:**

- The introduction of the Multimodal Latent Diffusion (MLD) method offers a new perspective on multi-modal generative modeling.
- The paper effectively tackles the coherence-quality tradeoff observed in existing multi-modal VAEs.
- The inclusion of experimental results provides concrete evidence for the paper's claims, underscoring its quality and relevance.

**Weaknesses:**

- Clarity in Presentation: While the paper is comprehensive, certain sections, especially those with plenty of mathematical formulations, might benefit from further simplification or more intuitive explanations for a broader audience.
- Dataset Diversity in Experiments: The current experiments focus primarily on simple or low-resolution datasets. Would the MLD approach's efficacy be consistent when tested on more popular, high-resolution datasets? Expanding the experimental evaluation to include such datasets might enhance the paper's applicability and appeal to the broader research community.
- Real-world Applications: The paper could be enriched by providing more real-world applications or use-cases to showcase the practical significance of MLD.

**Questions:**

From what I've gathered, the MLD approach involves concatenating latent vectors derived from several uni-modal autoencoders. Following this, a diffusion model is trained within this combined latent space. Subsequently, a mechanism is introduced to facilitate conditional generation. Based on my understanding:
1. How is the architecture of the autoencoders, such as the image AE and text AE, designed? In the context of stable diffusion, pure convolutional networks are utilized for both encoding and decoding. Does the MLD approach adopt a similar design for images?
2. How have you determined the latent dimensions for each modality, and what criteria influenced your decision on their dimensionality?
3. How scalable is the MLD approach when dealing with a large number of modalities or high-dimensional data within each modality?
4. Given the independent training of uni-modal autoencoders, how do you ensure that the concatenated latent space is cohesive and meaningful? Are there any challenges in ensuring convergence during training?
5. You mentioned you used 4 A100 GPUs for a total of roughly 4 months of experiments. Could you give more details about the training?
6. Does your implementation yield results that are in line with those presented in the original baseline studies?
7. The authors appear to have deviated from the official Style files and Templates as provided by the ICLR 2024 Call for Papers (https://iclr.cc/Conferences/2024/CallForPapers). Notably, there are discrepancies in the citation format.

---

> ### Author Response · Authors · 2023-11-21
>
> We thank the reviewer for acknowledging the main strengths of our work: a new angle to overcome the generative quality-coherence tradeoff, and the extensive experimental validation of our idea. We hope our thorough rebuttal will clarify the reviewers' doubts.
>
> > **Clarity in Presentation: While the paper is comprehensive, certain sections, especially those with plenty of mathematical formulations, might benefit from further simplification or more intuitive explanations for a broader audience**
>
> We do agree that the formulation might hinder accessibility of our work to a broader audience. However, we think that the mathematical rigor we adopt is a requirement for a venue like ICLR. To align with the reviewer’s request, we can commit to writing a new section in the Appendix called “Intuitive summary” which we hope will help the readers grasp at an intuitive level the key ideas of our method, without using the language of mathematics. Please refer to Appendix A.5 in the new version of the paper for a preliminary draft of such a discussion.
>
> > **Dataset Diversity in Experiments: The current experiments focus primarily on simple or low-resolution datasets. Would the MLD approach's efficacy be consistent when tested on more popular, high-resolution datasets? Expanding the experimental evaluation to include such datasets might enhance the paper's applicability and appeal to the broader research community.**
>
> In our research, we aim to demonstrate that the widely recognized concept of "latent fusion" through the product of experts, mixture of experts or combination thereof from the literature is not sufficient, particularly in terms of balancing generative quality and coherence. Our approach, which uses joint latent diffusion processes, effectively enables a new form of "latent fusion", whereby latent modalities jointly coevolve in the reverse, generative process. The datasets we selected for this study are not just adequate but crucial for drawing a meaningful comparison with the current state-of-the-art methods in "latent fusion".
>
> In light of a similar suggestion from Reviewer SqcE, we performed additional experiments (**see Appendix E.5**) with the **CelebAMask-HQ dataset** [Lee2020] as done in [Wesego2023], which consists of face images, each having a segmentation mask, and attributes, so 3 modalities. Our results are clear: the method we propose outperforms competitors from the state of the art, even with this new dataset.
> Other kinds of datasets, for example those including videos, are not appropriate for our goal because computing the coherence metric is not currently possible.
>
> * [Lee2020] Cheng-Han Lee, Ziwei Liu, Lingyun Wu, and Ping Luo. Maskgan: Towards diverse and interactive facial image manipulation. In IEEE Conference on Computer Vision and Pattern Recognition (CVPR), 2020.
> * [Wesego2023] Daniel Wesego and Amirmohammad Rooshenas, Score-Based Multimodal Autoencoders. https://arxiv.org/abs/2305.15708
>
> > **Real-world Applications: The paper could be enriched by providing more real-world applications or use-cases to showcase the practical significance of MLD.**
>
> In our exploration of the industrial applications of MLD, we have directed our focus toward the automotive industry, recognizing the potential impact of our research in this field. Specifically, we have conducted extensive experiments of MLD on automotive-related datasets. However, due to the private nature of these datasets, we are unable to share them publicly and have consequently omitted these specific tests from our paper.
>
> Despite this limitation, we can share insights into the practical use-cases we investigated. One key application of MLD in automotive technology is in mitigating the issues caused by faulty car sensors. In these applications, the text modality is absent, which hinders the use of many available pre-trained generative models (see also a comment by Reviewer 865d). Our model has shown promising results for in-car applications such as parking assistance, especially in situations where sensors are either faulty or their vision is occluded. These tests showcase the robustness and adaptability of MLD in handling real-world challenges in the automotive sector.
>
> **[Note: this reply continues in a separate comment.]**

---

> ### Author Response · Authors · 2023-11-21
> **.**
>
> >**How is the architecture of the autoencoders, such as the image AE and text AE, designed? In the context of stable diffusion, pure convolutional networks are utilized for both encoding and decoding. Does the MLD approach adopt a similar design for images?**
>
> In our MLD approach, we have emphasized the importance of tailoring each autoencoder to its specific input modality to achieve best performance. This principle is reflected in our design choices for image, audio, and text autoencoders. We apologize if references to supplementary material were not clear, but all details concerning the architectures are already reported in Appendix D. For the case of images, we consider the same convolutional architectures of the competitors, which combine convolutional and fully connected blocks. Additional details concerning other modalities are available in Appendix D.
>
> Note that our method is conceptually different from the Stable Diffusion (SD) approach, where pure convolutional networks are adopted. In that work, a “latent space” is only motivated by the need for reduced computational complexity of the diffusion model. In SD, the “latent” input to the diffusion model is a downsized image with 4 channels (details may vary depending on the SD version). In the VAE literature we are considering, latent space has a different meaning, and usually corresponds to a flattened vector which belongs to a real space of a given dimension. This difference has important practical implications: with our approach, it is much simpler to **mix** latent spaces of different modalities, whereas with SD such mixing would require careful engineering effort. Finally, recall that SD is unidirectional: text-to-image, but not the other way around.
>
> >**How have you determined the latent dimensions for each modality, and what criteria influenced your decision on their dimensionality?**
>
> This is a good question which is a research line per se! The manifold hypothesis [Gorban2018] tells us that natural data resides on a lower-dimensional space, and we tend to use small latent sizes, in accordance with what has been done in the literature. We have experiments with larger latent sizes and in general performance improves, up to a point. The interesting tradeoff in our case is that the smaller the latent size, the more computationally efficient is latent diffusion.
>
> [Gorban2018] Gorban AN, Tyukin IY. Blessing of dimensionality: mathematical foundations of the statistical physics of data. Philos Trans A Math Phys Eng Sci. 2018;376(2118):20170237. doi:10.1098/rsta.2017.0237
>
> >**How scalable is the MLD approach when dealing with a large number of modalities or high-dimensional data within each modality?**
>
> Our method MLD scales well in terms of the number of modalities, as well as in terms of the resolution of an individual modality in its input space. Concerning the number of modalities, this is visible for example in Fig. 18, Section E.3, where we vary the number of modalities used in the PolyMNIST dataset, and show that MLD performs extremely well, especially in terms of generation quality, while maintaining the highest coherence. Concerning resolution, this is a direct consequence of projecting the input in a lower-dimensional latent space.
>
> The primary bottleneck in MLD is the latent dimension selected for each modality. Larger latent sizes are necessary for complex modalities, and this affects the computational cost of our multi-time masked diffusion model. In reference to the previous remark, we select appropriate latent dimensions for each modality because we use well-studied datasets from the literature, for which our model does not require extremely high costs. For an application of MLD “in the wild”, a good engineering practice of studying the data first, for example through the lenses of our deterministic autoencoders, should be applied.
> Note that MLD provides an important advantage when compared to multimodal VAE-based approaches: there is no requirement for all latent spaces to have exactly the same dimension. This is evidently not true for e.g. Mixture based approaches, where the different modalities are required to have the same latent size (otherwise sum of latent means is not possible).
>
> **[Note: this reply continues in a separate comment.]**

---

> > ### Author Response · Authors · 2023-11-21
> > **.**
> >
> > >**Given the independent training of uni-modal autoencoders, how do you ensure that the concatenated latent space is cohesive and meaningful? Are there any challenges in ensuring convergence during training?**
> >
> > Thank you for the question, this is indeed a central element of our work. In alternative approaches, obtaining a cohesive latent space is a requirement. Different methods aim for the same result: to have geometrically **aligned** latent spaces for the different modalities. This is an implicit prerequisite to be able to mix the latent spaces into a unique one in a consistent way.
> >
> > In our work, we instead follow a different approach, as observed by the reviewer. In the first stage, separate autoencoders are trained independently. Consequently, there is no guarantee that the different latent spaces induced by the different modalities will share any form of geometric alignment. MLD is based on the concept of **concatenation** of the various latent spaces: there is no requirement for geometric cohesion. The overall joint distribution in such concatenated latent space is modeled through a diffusion process which exploits a **joint** score network, the fundamental tool which allows, starting from a set of independently sampled coordinates of gaussian noise, to mix the information and construct the necessary (non-linear) correlations.
> >
> > Concerning challenges during training, our approach is more stable than competitors. Indeed, as explained in the submitted version of the paper, we “ avoid any form of interference in the back-propagated gradients corresponding to the uni-modal reconstruction losses. Consequently, gradient conflict issues [Javaloy et al., 2022], where stronger modalities pollute weaker ones, are avoided.” (see Section 3, below Equation 2 in our paper).
> >
> > >**You mentioned you used 4 A100 GPUs for a total of roughly 4 months of experiments. Could you give more details about the training?**
> >
> > Our four-month long experimental campaign implements a comprehensive protocol that includes more than just training the MLD model. This time was allocated to training all competing models from scratch and conducting in-depth ablation studies. It's important to note that this duration represents the total time for the entire set of experiments, encompassing all aspects of our research, not just the training of MLD.
> >
> > >**Does your implementation yield results that are in line with those presented in the original baseline studies?**
> >
> > Yes, our extensive experimental validation report results which are extremely close to the original baselines.
> > ( Please see C2 and B.1.1  in  Hwang et al. (2021) ) (FID scores in table 1 in Palumbo et al. (2023) )
> >
> > * [Palumbo2023] E. Palumbo, et al. MMVAE+: Enhancing the Generative Quality of Multimodal VAEs Without Compromises, ICLR 2023, https://openreview.net/pdf?id=BYHy9WwxFU
> > * [Hwang et al. (2021 ] Hwang, HyeongJoo, et al. ]"Multi-view representation learning via total correlation objective." Advances in Neural Information Processing Systems 34 (2021): 12194-12207.
> >
> > >**The authors appear to have deviated from the official Style files and Templates as provided by the ICLR 2024 Call for Papers (https://iclr.cc/Conferences/2024/CallForPapers). Notably, there are discrepancies in the citation format.**
> >
> > We thank the reviewer for the suggestion, we updated the paper accordingly.

---

> > > ### Author Response · Authors · 2023-11-23
> > > **Any additional feedback?**
> > >
> > > Dear Reviewer yi7k,
> > >
> > > we were wondering if you had time to review our rebuttal, where we thoroughly answer all your questions.
> > >
> > > We improved our work, and we added experiments (see the impressive results in Appendix E.5) to address one of your questions regarding dataset diversity, by using CelebAMask-HQ, an high-resolution, three modality dataset.
> > >
> > > Could you please let us know if you are satisfied with the rebuttal? Thank you!

---

### Official Review · Reviewer_mRTE · 2023-11-01

**Soundness:** 3 good
**Presentation:** 2 fair
**Contribution:** 2 fair
**Rating:** 3
**Confidence:** 2

**Summary:**

this paper focuses on multi-modal image generation. the definite of multi-modal is multiple dataset distributions including text and image.  the proposed method is based on latent diffusion and the authors proposed som modification to make it work on multi-modal datasets.

**Strengths:**

* This paper focuses on classical problem in machine learning: multimodal dataset.
* the proposed method works better than other competing methods

**Weaknesses:**

* the results dont look very good. e.g., MLD painted birds in Fig 22.
* the paper lacks a overall diagram showing the whole model design. its' a bit difficult to understand the model design

**Questions:**

the other exciting methods seem quite weak. e.g., in Fig 20, MVAE cannot even generate digits very well, and in page 22, MVAE and MOPOE can't generate legible birds at all. are theses meaningful benchmark methods in 2023?
are there strong methods the author can use?

how is text generation done in the proposed method? I assume in figure 22, the models generated both images and text.

---

> ### Author Response · Authors · 2023-11-21
>
> >**Strengths:
> This paper focuses on classical problem in machine learning: multimodal dataset.
> the proposed method works better than other competing methods**
>
> We thank the reviewer for acknowledging that our work, which focuses on a classical and important problem in machine learning, i.e., multimodal generation, defined a new method that greatly outperforms all the considered alternatives from the state of the art.
>
> We have included additional experiments with the CelebAMask-HQ dataset [Lee2020] (see Appendix E.5 ), which consists of 3 modalities : face images, each having a segmentation mask, and attributes. Our MLD method outperforms competitors from the state of the art on both coherence and quality metrics.
>
> > **the other exciting methods seem quite weak. e.g., in Fig 20, MVAE cannot even generate digits very well, and in page 22, MVAE and MOPOE can't generate legible birds at all. are theses meaningful benchmark methods in 2023? are there strong methods the author can use?**
>
> The literature we target focuses on the problem of multimodal generative modeling, with a vast array of models that combine several input modalities in a joint, mixed latent space, which is then used to generate multimodal data. At the risk of being simplistic, we could state that the key differentiating factor in all such prior work is the method used to mix latent representations of different modalities into a unique latent space: broadly speaking, either the mixing happens according to a “product of experts”, a “mixture of experts”, or a combination thereof.
>
> The reviewer correctly acknowledges that the considered competitors suffer from limitations, especially from the perspective of generative quality. This is not a flaw of our comparative analysis, but a widely known literature result that pinpoints an open question: how can we achieve a better tradeoff in generative quality vs. coherence? This is *the current state of the art in 2022/2023*, as testified by works published in ICLR:
>
> * “MMVAE+: Enhancing the Generative Quality of Multimodal VAEs Without Compromises”, ICLR 2023, https://openreview.net/pdf?id=BYHy9WwxFU **(see fig. 7)**
> * “On the Limitations of Multimodal VAEs”, ICLR 2022, https://arxiv.org/pdf/2110.04121.pdf **(see Fig. 4)**
>
> The poor results we obtain for competitors is far from surprising, as we spent particular effort into replicating the experimental setup of our competitors, including architectures and hyper-parameters. Consequently, we claim that the answer to “are these meaningful benchmark methods in 2023?” is a sound yes.
>
> Importantly, the problems of our competitors are not related to architecture capacity, but to intrinsic limitations of the mixing method, which induces the generative quality/coherence tradeoff. This is exactly the conclusion reached in:
> “On the Limitations of Multimodal VAEs”, ICLR 2022, https://arxiv.org/pdf/2110.04121.pdf .
>
> > **the results dont look very good. e.g., MLD painted birds in Fig 22.**
>
> We respectfully disagree with this comment. It is clear that MLD outperforms competitors from an image quality generation point of view (see also the FID scores Table 16 Appendix E.4). This is a result of how our multi-time diffusion model achieves latent modality mixing, since the auto-encoder architectures we use are the same as for all competitors.
>
> > **how is text generation done in the proposed method? I assume in figure 22, the models generated both images and text.**
>
> Indeed, the methods we consider in this work can perform joint generation of all modalities. Text generation is carried out exactly like for all other modalities: a sampled latent variable is fed to a text-specific decoder.
>
> Note that our goal is not to engineer SOTA encoder-decoder architectures for the text modality, which would likely require using a LLM and several orders of magnitude more compute and training data. Instead, as clarified in the Appendix of the submitted paper, we adhere to related literature practice:  “For CUB, we use the same autoencoders architecture and implementation settings as in Daunhawer et al. [2022]. Laplace and one-hot categorical distributions are used to estimate likelihoods of the image and caption modalities respectively.”
>
> > **The paper lacks a overall diagram showing the whole model design. its' a bit difficult to understand the model design**
>
> We think the reviewer might have missed Fig. 4 in the Appendix of the submitted paper, where we clearly display a diagram of the model design.

---

> > ### Author Response · Authors · 2023-11-23
> > **Any additional feedback?**
> >
> > Dear Reviewer mRTE,
> > we hope you had time to review our rebuttal, where we answer thoroughly to all your questions.
> >
> > Since one of the remarks was about image quality, we would like to invite the reviewer to have a look at the new Appendix E.5, where we have included additional experiments with CelebAMask-HQ, an high-resolution dataset with three modalities, in which we show impressive results obtained by MLD.
> >
> > Could you please let us know if you are satisfied with the rebuttal? Thank you!

---

### Author Response · Authors · 2023-11-21
**Message to the reviewers**

Dear Reviewers,
We sincerely appreciate the time and effort you have dedicated to reviewing our paper. Your comments and questions have been addressed with care and dedication in our revision:
* We have answered all your questions, did our best to clarify doubts, and amended our paper with such clarifications (modifications to the main, plus Appendix A.5). This includes also the  release of the anonymized code.
* We performed additional experiments that were requested, to include latest state of the art methods as of MMVAE+ ICLR 2023 (main paper, Table 1, 2, and 3, Fig 2,Appendix Table 11,12,13,14,15,16, Fig 22)
* We performed additional experiments (Appendix E.5) that were requested, to illustrate the performance of our method using a larger and higher-resolution multimodal dataset such as **CelebAMask-HQ**, and compared our results with competitors.
Thank you! All these new experiments further extend the breath of our work, and are beneficial to its aesthetic (**please, check qualitative results in Appendix E.5!**). Also, these new results did not change the conclusion: our method outperforms other works, even concurrent submissions to ours. Please, refer to the table below:

| Models                | Attributes           |                 | Image                |                      |                      |                      | Mask      |           |
| :-------------------: | :------------------: | :-------: | :------------------: | :------------------: | :------------------: | :------------------: | :-------: | :-------: |
|                       | Img + Mask           | Img       | Att + Mask           | Mask                 | Att                  | Joint                | Img+Att   | Img       |
|                       | F1                   | F1        | FID                  | FID                  | FID                  | FID                  | F1        | F1        |        |
|    (Wesego2023)                 |                      |           |                      |                      |                      |                      |           |           |
| SBM-RAE               | 0\.62                | 0\.6      | 84\.9                | 86\.4                | 85\.6                | 84\.2                | 0\.83     | 0\.82     |
| SBM-RAE-C             | 0\.66                | 0\.64     | 83\.6                | 82\.8                | 83\.1                | 84\.2                | 0\.83     | 0\.82     |
| SBM-VAE               | 0\.62                | 0\.58     | 81\.6                | 81\.9                | 78\.7                | 79\.1                | 0\.83     | 0\.83     |
| SBM-VAE-C             | 0\.69                | 0\.66     | 82\.4                | 81\.7                | 76\.3                | 79\.1                | 0\.84     | 0\.84     |
| MOPOE                 | 0\.68                | **0\.71** | 114\.9               | 101\.1               | 186\.8               | 164\.8               | 0\.85     | **0\.92** |
| MVTCAE                | 0\.71                | 0\.69     | 94                   | 84\.2                | 87\.2                | 162\.2               | **0\.89** | 0\.89     |
| MMVAE+                | 0\.64                | 0\.61     | 133                  | 97\.3                | 153                  | 103\.7               | 0\.82     | 0\.89     |
| **MLD (ours)**      | **0\.72**|0\.69| **52\.75** | **51\.73**|**53\.09** |**54\.27** | 0\.87     | 0\.87     |

We have noticed a general trend in your reviews, which is probably rooted in the fact that, in the past few months, multimodality has landed into text-based applications, including conversational agents and applications in which text is de facto a “master modality”. Some recent work, that are concurrent or close to concurrent to our submission, exploit huge computational power to deal with billion-sample scale multimodal datasets. These beautiful engineering artifacts are impressive and are all the rage on conference proceedings and social media. However, we have deliberately focused on the line of research that aims at multimodal latent fusion, which has received a lot of attention starting from 2016, with several articles that appeared in NeurIPS, ICML and with the current state of the art that appeared at ICLR 2022 and ICLR 2023.

Despite reviewers trying to steer our experimental campaign toward such endeavor, we believe that our quest for a deeper understanding of how mixing modalities into a useful latent representation that allows joint and conditional generation, shedding lights on an open question in a well-established literature, and proposing an effective approach to solve it, is as important as showing attractive high resolution images or videos.

We trust that our rebuttal has addressed all of your questions, with concrete explanations, and tangible results. We hope that you will be satisfied with our updated work, and that you will lean toward accepting our paper for being discussed at ICLR 2024!

---

### Meta-Review · Area_Chair_ydWT · 2023-12-14

**Metareview:**

The paper introduces a novel approach for multi-modal generative modeling, addressing the coherence-quality tradeoff in Variational Autoencoders (VAEs) by employing independently trained, deterministic uni-modal autoencoders and a masked diffusion model for generative modeling. Reviewer mRTE rated the paper as reject with a low-quality review and did not engage in the discussion after the author rebuttal. AC discounted this review comment in the final decision. Reviewer yi7k had a positive view, appreciating the paper's tackling of the coherence-quality tradeoff and the experimental results. They raised concerns about the clarity of presentation, dataset diversity, and real-world applications. The authors addressed these points in their rebuttal. Reviewer 865d rated the submission marginally below acceptance, noting the lack of comparison with purely diffusion-based models and limitations in dataset diversity. The authors responded with an explanation for not comparing with CoDi and additional dataset experiments and acknowledged the scalability issue for the proposed framework. 865d kept the rating. Reviewer SqcE pointed out issues with the methodology, lack of theoretical backing, and concerns over the model's flexibility and scalability. The authors' rebuttal included further clarification. The reviewer increased the rating to 6.
Overall, the reviewers' responses lack enthusiasm, and their collective assessment leans towards a borderline negative rating. After careful consideration of the paper, the reviewers' comments and the author response, the AC decides to recommend reject (see "Justification For Why Not Higher Score" for further suggestions).

**Justification For Why Not Higher Score:**

Overall, the reviewers' responses lack enthusiasm, and their collective assessment leans towards a borderline negative rating. The paper introduces an interesting method and demonstrates some promising potentials. However, the Area Chair believes that in its current form, the paper falls short of the acceptance criteria and requires additional revision. It's important to enhance the positioning of the proposed framework within the context of recent advancements in generative models. This enhancement could be achieved by offering a more thorough comparison with relevant studies, improving the clarity of the presentation, and providing stronger theoretical support. The Area Chair also suggests that the authors might benefit from selecting a more precise and distinct title, one that sets their work apart in the extensive field of generative modeling and multi-modal learning.

**Justification For Why Not Lower Score:**

N/A

---

### Decision · Program_Chairs · 2024-01-16

Reject